# Hypoxia-induced conversion of sensory Schwann cells into repair cells is regulated by HDAC8

Nadège Hertzog [1,2], Mert Duman [1,2,5], Maëlle Bochud [2,5], Valérie Brügger-Verdon [2,5], Maren Gerhards [1], Felicia Schön [1], Franka Dorndecker[1], Dies Meijer [3], Robert Fledrich [4], Ruth Stassart [4], Devanarayanan Siva Sankar [2], Jörn Dengjel [2], Sofía Raigón López [1] & Claire Jacob [1,2] ✉

After a peripheral nerve injury, Schwann cells (SCs), the myelinating glia of the peripheral nervous system, convert into repair cells that foster axonal regrowth, and then remyelinate or re-ensheath regenerated axons, thereby ensuring functional recovery. The efficiency of this mechanism depends however on the time needed for axons to regrow. Here, we show that ablation of histone deacetylase 8 (HDAC8) in SCs accelerates the regrowth of sensory axons and sensory function recovery. We found that HDAC8 is specifically expressed in sensory SCs and regulates the E3 ubiquitin ligase TRAF7, which destabilizes hypoxia-inducible factor 1-alpha (HIF1α) and counteracts the phosphorylation and upregulation of c-Jun, a major inducer of the repair SC phenotype. Our study indicates that this phenotype switch is regulated by different mechanisms in sensory and motor SCs and is accelerated by HDAC8 downregulation, which promotes sensory axon regeneration and sensory function recovery.

After a traumatic injury, the peripheral nervous system (PNS) can efficiently regenerate. This is to a large extent due to the high plasticity of Schwann cells (SCs) (reviewed in ref. 1), the myelinating glia and the main glial cell type of the PNS. Indeed, SCs react rapidly to a traumatic injury of the PNS by actively demyelinating and converting into repair cells that secrete neurotrophic factors which promote axonal regrowth, and form bands of Büngner that guide axons back to their former target[2]. SCs then remyelinate or re-ensheath regenerated axons, which leads to successful functional recovery[1,3]. In some cases, however, when the injury has created a large gap between axons and their target or in aged individuals, target re-innervation and functional recovery can partially or completely fail[4]. Understanding the molecular mechanisms that control SC plasticity after injury has the potential to

reveal drug targets for increasing SC plasticity and thereby accelerating axonal regrowth and remyelination efficiency.

The transcription factor c-Jun, which is rapidly upregulated in SCs after a PNS injury, is widely recognized as a master inducer of the SC demyelination process and conversion into repair cells, and is critical for axonal regrowth after injury[5–8]. Interestingly, c-Jun expression levels in SCs after a PNS lesion are lower in aged individuals compared to young adults, and restoring c-Jun levels in aged individuals leads to axonal regeneration of similar efficiency as in young adults[9]. Although the functions and target genes of c-Jun have been extensively studied in SCs after injury, the mechanisms that control c-Jun upregulation remain partially understood. Wagstaff et al.[9] showed an involvement of Sonic Hedgehog signaling in maintaining high c-Jun levels at 7 days

[1]Institute of Developmental Biology and Neurobiology, Faculty of Biology, Johannes Gutenberg University Mainz, Mainz, Germany. [2]Department of Biology, University of Fribourg, Fribourg, Switzerland. [3]Center for Discovery Brain Sciences, University of Edinburgh, Edinburgh, UK. [4]Paul Flechsig Institute, Center of Neuropathology and Brain Sciences, University of Leipzig, Leipzig, Germany. [5]These authors contributed equally: Mert Duman, Maëlle Bochud, Valérie Brügger-Verdon. ✉e-mail: cjacob@uni-mainz.de

post sciatic nerve cut lesion, but not at earlier time points. mTORC1 signaling is reactivated in SCs after lesion and has been shown to ensure proper c-Jun upregulation in SCs after injury by promoting c-Jun translation[10]. In addition, O-GlcNAcylation of c-Jun by the enzyme O-GlcNAc transferase attenuates c-Jun phosphorylation and activity, and allows timely remyelination[11]. Indeed, persistent c-Jun activity impairs SC redifferentiation and axon remyelination[7,11]. We have previously shown that the two highly homologous class I histone deacetylases 1 and 2 (HDAC1/2) are critical in SCs for the remyelination process after a PNS injury[12,13]. However, we also showed that HDAC2 is upregulated too early in SCs after injury, already 1 day after a sciatic nerve crush lesion. Indeed, HDAC2 upregulation following injury leads to precocious upregulation of the transcription factor Oct6, which in turn counteracts c-Jun upregulation in SCs, thereby slowing down axonal regrowth. We also demonstrated that a short treatment with an HDAC1/2 inhibitor following a PNS injury accelerates axonal regrowth and functional recovery without impairing remyelination[12]. These findings indicate that the regeneration process after a PNS injury is not optimal and can be improved. Other members of the HDAC family have also been described to be involved in the regeneration process of the PNS after injury[14–16]. Indeed, HDAC3, another class I HDAC, has been reported to prevent precocious remyelination after lesion[17], while members of class II HDACs have been shown to be involved in the remyelination process[18].

A PNS lesion abruptly interrupts oxygen supply and therefore creates a hypoxic environment[19]. Hypoxia is known as a strong inducer of c-Jun phosphorylation and upregulation[20–22]. Indeed, hypoxia-inducible factor 1-alpha (HIF1α), a transcription factor that is rapidly targeted to the proteasome and degraded in the cell cytoplasm under normoxic conditions, is instead rapidly upregulated in hypoxic conditions by inhibition of its degradation (reviewed in ref. 23). Under hypoxia, HIF1α upregulation can promote c-Jun-N-terminal kinases (JNK) activation[21,24] and both HIF1α and phosphorylated JNK translocate to the cell nucleus[23,25,26], which leads to JNK-dependent c-Jun phosphorylation and to the activation of *c-Jun* transcription[20,21] and c-Jun stabilization[27]. The mechanisms regulating HIF1α stabilization and degradation have been extensively studied[23,28], particularly in cancer cells[29–31]. However, they remain partially understood and little is known about HIF1α regulation in SCs. Here, we show that HDAC8, a member of the class I HDAC family of proteins whose functions in SCs remained so far unreported, counteracts the stabilization of HIF1α in SCs, thereby slowing down HIF1α-dependent c-Jun phosphorylation and upregulation, axonal regrowth, and functional recovery after injury. Unexpectedly, we found that HDAC8 regulates this mechanism specifically in SCs ensheathing sensory axons, thereby specifically controlling the regrowth of sensory axons and recovery of the sensory function.

## Results

### HDAC8 ablation promotes sensory axons regrowth and sensory function recovery

We found that after a sciatic nerve crush lesion, HDAC8 expression is upregulated in peripheral nerves distal to the lesion site (Fig. 1a) and is exclusively or in major part found in the cell cytoplasmic compartment, until at least 12 days post lesion (dpl) (Fig. 1b). To test for a potential function of HDAC8 in SCs during the regeneration process after lesion, we ablated HDAC8 specifically in adult SCs. To this aim, we crossed *Hdac8* floxed mice with mice expressing the tamoxifen-inducible CreERT2 recombinase under control of the *Plp* promoter (HDAC8 KO) that leads to recombination in SCs (and in oligodendrocytes and satellite glia, but not in adult neurons or other cells of the sciatic nerve[32–38]) and used *Plp*-CreERT2-negative littermates as control mice. To induce HDAC8 ablation, we injected 2 mg tamoxifen per day for five consecutive days. Control mice received tamoxifen injections simultaneously to HDAC8 KO mice. Mice were submitted to a sciatic

nerve crush lesion two weeks after tamoxifen injections, where HDAC8 was efficiently ablated in SCs (Supplementary Fig. 1a). Surprisingly, the regeneration process after lesion was not impaired by the ablation of HDAC8, but instead was promoted. Indeed, we found by electron microscopy less intact myelin rings at 5 dpl and less degenerated myelin rings at 12 dpl in HDAC8 KO compared to control nerves (Fig. 1c, d), suggesting improved demyelination and clearance of myelin debris in the absence of HDAC8 in SCs. In addition, more axons per surface area, including small-caliber axons (<2 μm diameter) in regenerating axonal clusters and larger-caliber axons (≥2 μm diameter), had regrown at 12 dpl (Fig. 1d) in HDAC8 KO compared to control mice. To test for a potential improvement in functional recovery, we crossed *Hdac8* floxed mice with mice expressing the tamoxifen-inducible CreERT2 recombinase under control of the *P0* promoter (also called thereafter HDAC8 KO, only used for functional recovery experiments) that recombines only SCs (and not oligodendrocytes or other cells of the PNS or CNS[32,39–41], and therefore ensures that the phenotype observed is only due to ablation of HDAC8 in SCs) and used *P0*-CreERT2-negative littermates as control mice. The *P0*-CreERT2 line, however, leads to recombination in myelinating Schwann cells only (Supplementary Fig. 2 and refs. 39,41), therefore we used the *Plp*-CreERT2 line, which recombines all SCs including myelinating and non-myelinating SCs (Supplementary Fig. 2 and refs. 33,39), for morphological, ultrastructural and molecular analyses. *P0*-CreERT2;*Hdac8* fl/fl and control littermate mice were submitted to a sciatic nerve crush lesion two weeks after tamoxifen injections, where HDAC8 was also efficiently ablated (Supplementary Fig. 1b). We found that sensory function recovery analyzed by Toe pinch and von Frey tests was faster in HDAC8 KO mice (Fig. 1e and Supplementary Fig. 3) and correlated with hind paw re-innervation (Supplementary Fig. 4). However, motor function recovery analyzed by Rotarod and Inverted grid tests (Fig. 1e) and remyelination (Supplementary Fig. 5a, b) were comparable in both HDAC8 KO and control mice and we did not detect compensatory expression of HDAC1, HDAC2 or HDAC3 (Supplementary Fig. 5b), the other members of class I HDACs. Contralateral nerves at all time points analyzed (5, 12, 30 dpl) did not show any defect in myelin stability (no degenerated myelin ring), number of myelinated axons (and no demyelinated/unmyelinated axon in 1-to-1 relationship with a SC was found), and Remak axons or g-ratio (Supplementary Fig. 6). In addition, myelin maintenance without lesion using *P0*CreERT2-*Hdac8* fl/fl and control littermate mice was not affected until at least 6 months after tamoxifen injections (Supplementary Fig. 7) and HDAC8 KO mice were indistinguishable from their control littermates until at least 6 months after tamoxifen injections. HDAC8 ablation in developing SCs using the *Dhh*Cre mouse line[42] where recombination occurs at embryonic day 13.5 (*Dhh*Cre;*Hdac8* fl/fl and Cre-negative littermate mice used as controls) also did not lead to detectable myelination defects at the ultrastructural and protein levels (except for a transient non-significant trend for increased P0 levels at P21), or to detectable compensatory expression of the other members of class I HDACs, except for a transient moderate decrease of HDAC2 levels at P10 (Supplementary Fig. 8), *Dhh*Cre;*Hdac8* fl/fl mice were indistinguishable from their control littermates, and showed similar life expectancy and breeding efficiency compared to control littermates. Our results indicate that HDAC8 is mostly dispensable in SCs for myelin development and maintenance, but do not fully exclude potential compensatory mechanisms or defects occurring at different time points, which would require further analyses.

To continue our analyses of HDAC8 functions in SCs during the regeneration process after sciatic nerve crush lesion, we carried out whole-nerve stainings for Stathmin-2 (STMN2, also called SCG10) and growth-associated protein 43 (GAP43) in sciatic nerves of HDAC8 KO and control mice (*Plp*-CreERT2;*Hdac8* fl/fl mice and Cre-negative control littermate mice) at 3 dpl. STMN2 is a highly selective marker of regrowing axons concentrating at the axon growth cone after sciatic

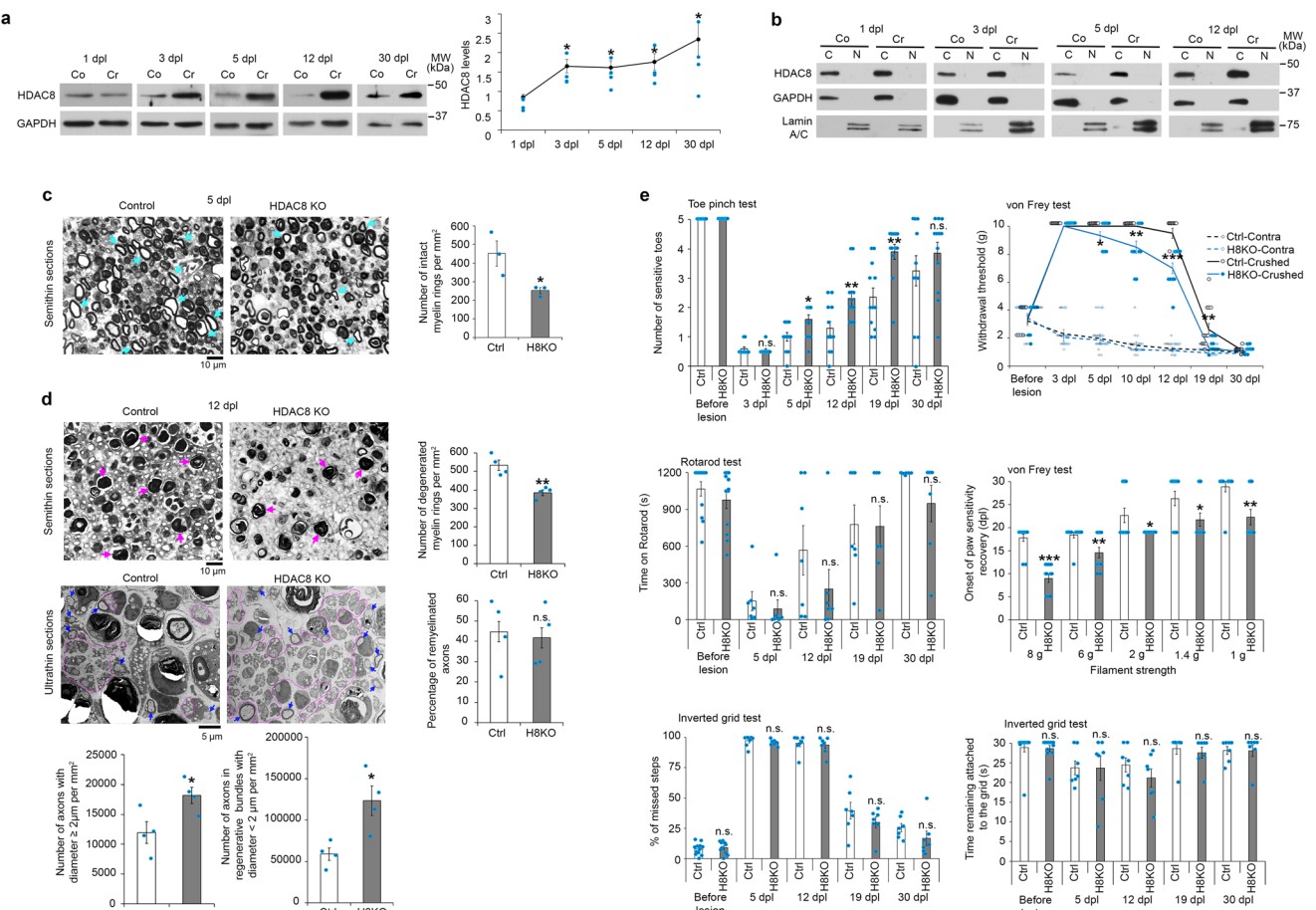

**Fig. 1 | HDAC8 ablation in SCs promotes sensory function recovery. a, b** HDAC8 Western blot on wild type (WT) adult mouse (**a**) whole sciatic nerve lysates or (**b**) after subcellular fractionation of cytoplasmic (C) and nuclear (N) fractions and quantification normalized to GAPDH (**a**), at 1, 3, 5, 12 and 30 dpl in crushed (Cr) compared to contralateral (Co) sciatic nerves. (**b**) GAPDH and Lamin A/C = markers of cytoplasmic and nuclear fractions, respectively. (**a**) $N = 4$ (1, 3, 5, 12 dpl) or 5 (30 dpl) animals per group, (**b**) 3 WT mice per time point, representative images are shown. **c** Semithin cross-sections of HDAC8 KO (H8KO) and Control (Ctrl) adult mouse sciatic nerves at 5 dpl and graphs showing the number of intact myelin rings per mm$^2$ (3 sections per animal). $N = 3$ animals per group. Turquoise arrows = intact myelin rings. **d** Semithin (upper panel) and ultrathin (lower panel) cross-sections of H8KO and Ctrl adult mouse sciatic nerves at 12 dpl and graphs showing the number of degenerated myelin rings per mm$^2$, the percentage of remyelinated axons, the

number of axons with diameter ≥2 μm per mm$^2$, and the number of axons in regenerating clusters (diameter <2 μm per mm$^2$). $N = 4$ animals per group (3 semithin sections per animal, 5 images of 3 ultrathin sections per animal). Upper panel: magenta arrows = degenerated myelin rings. Lower panel: blue arrows = remyelinated axons, areas delineated by pink line = regenerating axon clusters. **e** Graphs showing the performance of H8KO and Ctrl mice at the Toe pinch, von Frey, Rotarod and Inverted grid tests (total number of steps per animal: 17 to 136) before or after sciatic nerve crush lesion. $N$ (Toe pinch, von Frey) = 12, n (Rotarod, Inverted grid) = 7 (after lesion) or 12 (before lesion) animals per group. Paired (**a**) or unpaired (**c, d, e**) two-tailed (black asterisks) or one-tailed (n.s.) Student's t-tests or two-way mixed ANOVA followed by post-hoc t-tests (von Frey), p values: *<0.05, **<0.01, ***<0.001, n.s. = non-significant, values = mean, error bars = s.e.m. Source data are provided as a Source data file.

nerve injury (however, also expressed in uninjured axons[43,44], see also Supplementary Fig. 9) that preferentially labels sensory axons, while GAP43 is a general marker of regrowing axons[44]. Consistent with faster sensory function recovery, we found that at 3 dpl STMN2-labeled axons had regrown over a longer distance and with higher density up to 2.5 mm downstream the lesion site in sciatic nerves of HDAC8 KO mice compared to control mice (Fig. 2a), whereas we could not detect a significant difference of axonal length for GAP43-labeled axons, which we found in only mild increased density at 1.5 mm downstream the lesion site (Fig. 2b). We thus asked whether HDAC8 is specifically expressed in SCs associated with sensory axons, which could potentially explain why HDAC8 ablation has a specific effect on the regrowth and functional recovery of sensory axons. To answer this question, we co-stained HDAC8 with STMN2 that preferentially labels sensory axons and Neurofilament that labels all axons. Indeed, a large majority of SCs expressing high levels of HDAC8 (96 ± 3%) were associated with STMN2-positive axons, whereas a very low fraction (4 ± 3%) of SCs associated with STMN2-negative axons (Neurofilament-positive)

expressed HDAC8 (Fig. 2c). To strengthen this finding, we tested the expression of HDAC8 in ventral roots, which consist mostly in motor axons, and in sural nerves, which consists exclusively in sensory axons. Indeed, HDAC8 was not detected in any SC present in ventral roots but was abundantly expressed in SCs of sural nerves (Supplementary Fig. 10). Of note, HDAC8 was not expressed in all SCs surrounding sensory axons but was expressed in around 90% of SCs surrounding Isolectin B4 (marker of Remak axons)-positive axons and in around 60% of SCs surrounding myelinated Stathmin-2-positive axons (Supplementary Fig. 2). Overall 54 ± 6% of cells (detected by their nucleus) present in adult mouse sciatic nerves expressed HDAC8 ($n = 3$ adult mice, 82 to 207 nuclei counted per mouse). HDAC8 was also expressed in subsets of axons (Fig. 2c and Supplementary Fig. 10). Whether SCs expressing HDAC8 specifically surround A-alpha, A-beta, and/or A-delta myelinated axons remains to be determined, but this analysis will require to identify specific markers for each of these sensory myelinated axon subtypes. Analysis of macrophage numbers indicated no difference between sciatic nerves of HDAC8 KO and control

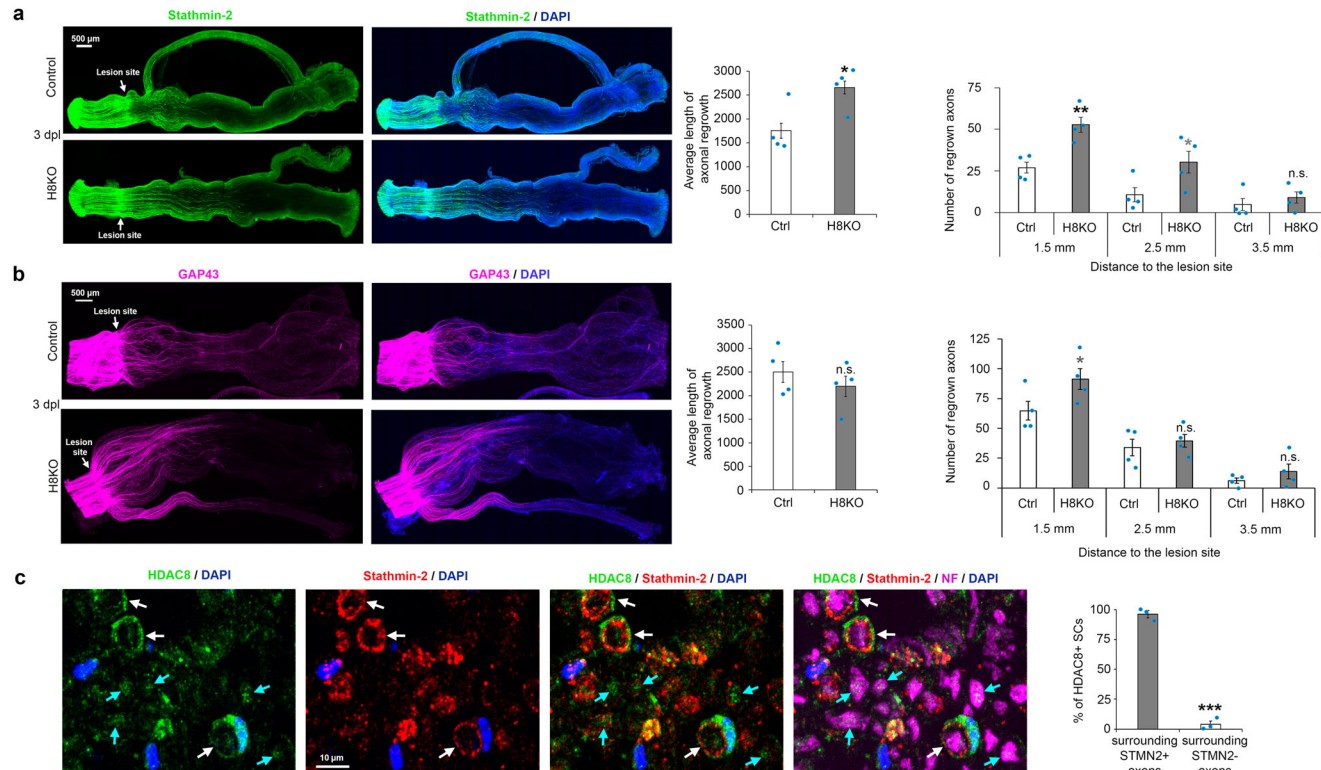

**Fig. 2 | HDAC8 ablation in SCs promotes sensory axon regeneration. a, b** Whole sciatic nerve immunofluorescence (IF) of (**a**) Stathmin-2 (green) or (**b**) GAP43 (magenta), and DAPI labeling (blue, nuclei) and quantification of axonal regrowth at 3 dpl in HDAC8 KO (H8KO) nerves compared to Control (Ctrl) nerves. $N = 4$ animals per group (average length: 43 to 79 regrowing axons per animal). **c** Co-immunofluorescence (z-stacks projetions) of HDAC8 (green), Stathmin-2 (red) and Neurofilament (NF, magenta) and DAPI labeling (blue) in sciatic nerve cross-sections of wild type mice at 1 dpl (proximal part). Sections of 3 animals analyzed, a representative image is shown. White arrows = HDAC8-positive SCs surrounding Stathmin-2-positive axons, blue arrows = HDAC8-positive axons (NF-positive). The graph shows the percentage of HDAC8-positive SCs surrounding Stathmin-2-positive and Stathmin-2-negative axons. $N = 3$ animals per group, 25 to 94 HDAC8-positive SCs per animal. Unpaired two-tailed (black asterisks) or one-tailed (gray asterisks or n.s.) Student's t-tests, $p$ values: *<0.05, **<0.01, ***<0.001, n.s. = non-significant, values = mean, error bars = s.e.m. Source data are provided as a Source data file.

littermate mice at 1 or 5 dpl (Supplementary Fig. 11). Taken together, our results identify HDAC8 as a specific marker of SCs associated with a large fraction of sensory axons in adult sciatic nerves and show that the ablation of HDAC8 in SCs promotes myelin debris clearance, regrowth of sensory axons and sensory function recovery.

## HDAC8 ablation enhances hypoxia-induced c-Jun phosphorylation and upregulation

Molecularly, we found that the levels of c-Jun (Fig. 3a, b) and phosphorylated c-Jun (Fig. 3b) were increased in SCs of HDAC8 KO compared to control mice, already at 1 dpl. c-Jun expression levels remained increased in SCs of HDAC8 KO mice until at least 3 dpl and returned to control levels at 5 dpl (Fig. 3a). Although Stathmin-2 expression is rapidly decreased distal to the lesion side[43,44], we could still detect remnants of Stathmin-2 signal around axons, which we used to ask whether c-Jun levels were increased in SCs in contact with sensory axons. Indeed, c-Jun levels were robustly increased in SCs surrounding Stathmin-2-positive axons (Supplementary Fig. 12). Oct6 expression levels in SCs were not affected by the ablation of HDAC8 (Supplementary Fig. 13), indicating that the mechanisms underlying c-Jun upregulation in HDAC8 KO SCs are different than the mechanisms responsible for c-Jun upregulation in HDAC1/2 KO SCs[12].

To elucidate these mechanisms, we used purified primary rat SCs cultured under conditions that mimic the conversion into the repair SC phenotype occurring after a peripheral nerve lesion, where myelin proteins are downregulated and c-Jun upregulated[12,13,45]. To downregulate HDAC8 in primary SCs, we used a lentiviral vector carrying a highly efficient HDAC8-specific shRNA (Fig. 3c). HDAC8 downregulation in SCs led to increased levels of c-Jun and phosphorylated c-Jun compared to levels in SCs transduced with a lentiviral vector carrying a non-targeting control shRNA (Supplementary Fig. 14a), an effect that was, however, more significant under hypoxic conditions, either by incubation in a hypoxia chamber (Fig. 3d, e) or incubation with cobalt chloride ($CoCl_2$) which mimics hypoxic conditions (Fig. 3f, g). Hypoxia is known as a robust inducer of c-Jun expression and phosphorylation, and occurs after a peripheral nerve injury[19–22,46,47]. Additionally, knocking out HIF1α specifically in adult SCs by crossing *Plp*-CreERT2 mice with *Hif1α*-floxed mice and inducing *Hif1α* recombination by tamoxifen injections, such as described above for *Hdac8* recombination, leads to reduced c-Jun upregulation after lesion (Supplementary Fig. 15). For these reasons, we carried out our cell cultures under hypoxia and tested whether hypoxia is involved in HDAC8-mediated regulation of c-Jun expression and phosphorylation after injury. In addition to increased c-Jun protein levels, we found that HDAC8 downregulation in primary SCs led to increased *c-Jun* mRNA levels under hypoxic conditions (Fig. 3h), but only to a trend under normoxia (Supplementary Fig. 14b). Consistently, HDAC8 downregulation resulted in increased activation of the *c-Jun* promoter under hypoxic conditions (Fig. 3i), but not under normoxia (Supplementary Fig. 14c). These results indicate that HDAC8 counteracts c-Jun upregulation and phosphorylation, in particular under hypoxic conditions.

Hypoxia leads to the stabilization of HIF1α and its translocation to the nuclear compartment (ref. 23 and Supplementary Fig. 16). Consistent with HDAC8 cytoplasmic localization in contralateral and

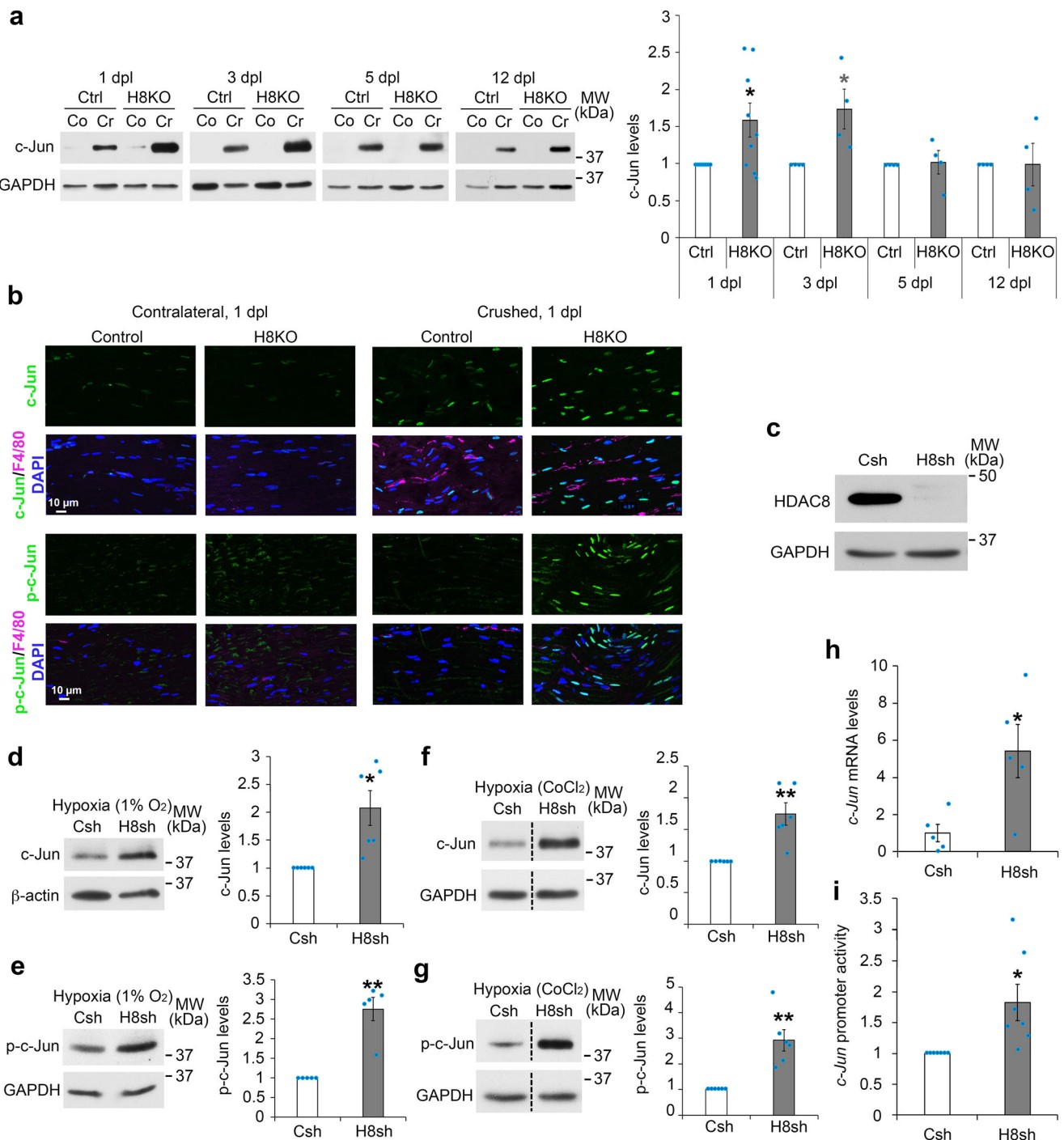

**Fig. 3 | Ablation of HDAC8 in SCS leads to increased c-Jun and phospho-c-Jun levels. a** c-Jun Western blot and quantification normalized to GAPDH at 1, 3, 5 and 12 dpl in crushed sciatic nerves of HDAC8 KO compared to Control mice. $N = 9$ (1 dpl) or 4 (3, 5, and 12 dpl) animals per group. **b** Co-immunofluorescence of c-Jun (green, upper panels) or phospho-c-Jun (p-c-Jun, green, lower panels) and F4/80 (magenta, macrophage marker), and DAPI labeling (blue, nuclei) in longitudinal sections of crushed and contralateral HDAC8 KO and Control sciatic nerves at 1 dpl. Three animals per group, representative images are shown. **c** HDAC8 and GAPDH (loading control) Western blots on lysates of rat SCs transduced with lentiviruses carrying an HDAC8-specific or a non-targeting control shRNA. Multiple (more than 3 times) independent experiments, representative images are shown. **d–g** Western blot of c-Jun (**d, f**) or phospho-c-Jun (**e, g**) in primary rat SCs cultured under conditions mimicking the conversion into the repair phenotype in hypoxia (**d, e**: in hypoxic chamber for 16 h; **f, g**: CoCl₂ for 16 h), and quantification normalized to GAPDH in cells incubated with lentiviruses carrying an HDAC8 shRNA (H8sh) or control shRNA (Csh). Dashed lines indicate that samples were run on the same gel but not on consecutive lanes. $N = 6$ (**d, f, g**) or 5 (**e**) independent experiments per group. **h** Quantification of *c-Jun* mRNA levels by qRT-PCR in primary rat SCs cultured as above (**f, g**) and incubated with lentiviruses carrying an H8sh or Csh. $N = 5$ independent experiments per group. **i** Quantification of *c-Jun* promoter activity by luciferase gene reporter assay in cells cultured as above (**f, g, h**) and incubated with lentiviruses carrying an H8sh or Csh. $N = 7$ independent experiments per group. Paired (**a, d–g, i**) or unpaired (**h**) two-tailed (black asterisks) or one-tailed (gray asterisks). Student's t-tests, *p* value: *<0.05, **<0.01, values = mean, error bars = s.e.m. Source data are provided as a Source data file.

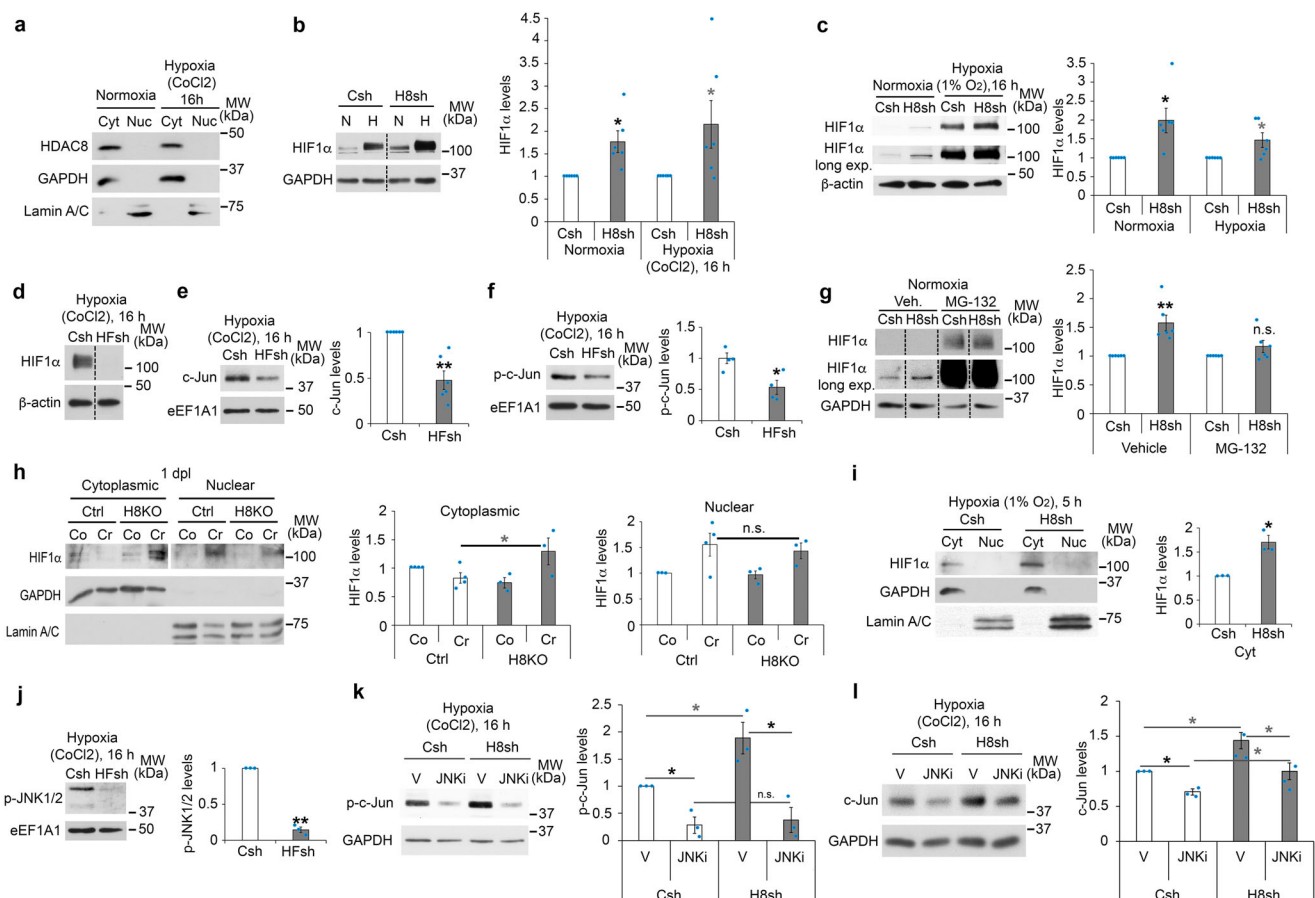

**Fig. 4 | HDAC8 ablation activates a HIF1α/JNK/c-Jun axis. a**, **h**, **i** Western blot of HDAC8 (**a**) or HIF1α (**h**, **i**) on lysates of rat SCs cultured in conditions mimicking the conversion into the repair phenotype under normoxia or hypoxia (**a**, CoCl$_2$ for 16 h; **i**, hypoxic chamber for 5 h) or on lysates of contralateral (Co) and crushed (Cr) sciatic nerves of HDAC8 KO (H8KO) and control (Ctrl) mice at 1 dpl (**h**), after subcellular fractionation (Cyt = cytoplasmic, Nuc = nuclear) and quantification of the nuclear fraction normalized to Lamin A/C (**h**) or of the cytoplasmic fraction normalized to GAPDH (**h**, **i**) in cells where HDAC8 was downregulated by shRNA (H8sh) compared to control shRNA (Csh) (**i**), or in H8KO compared to Ctrl nerves (**h**). **b**–**g**, **j** HIF1α (**b**–**d**, **g**), c-Jun (**e**), phospho-c-Jun (p-c-Jun, **f**) and phospho-JNK1/2 (p-JNK1/2, **j**) Western blot on lysates of rat SCs cultured as above (**a**) in normoxia (N, **b**, **c**, **g**) or hypoxia (H, 16 h (**b**, **d**–**f**, **j**) with CoCl$_2$ or (**c**) in hypoxia chamber (**c**), and (**g**)

incubated with the proteasome inhibitor MG-132 or its vehicle for 4 h, and quantification normalized to GAPDH, β-actin or eEF1A1 in cells incubated with lentiviruses carrying H8sh (**b**, **c**, **g**) or a HIF1α shRNA (HFsh, **d**–**f**, **j**) or Csh. **k**, **l** Phospho-c-Jun (p-c-Jun, **k**) or c-Jun (**l**) Western blots on lysates of rat SCs cultured as above (**a**) in hypoxia and incubated with H8sh or Csh lentivirus, and with a JNK inhibitor (JNKi) or its vehicle (V), and quantification normalized to GAPDH. Dashed lines=samples run on the same gel but not on consecutive lanes. Three (**a**) or 6 (**d**) independent experiments, representative images are shown. **h** N = 4 (Ctrl) or 3 (H8KO) animals. N = 3 (**i**, **j**, **k**, **l**), 4 (**f**) or 6 (**b**, **c**, **e**, **g**) independent experiments. Paired (**b**, **c**, **e**, **g**, **i**, **j**, **k**, **l**) or unpaired (**f**, **h**) two-tailed (black asterisks) or one-tailed (gray asterisks, n.s.) Student's t-tests, p values: *<0.05, **<0.01, n.s. = non-significant, values = mean, error bars = s.e.m. Source data are provided as a Source data file.

crushed sciatic nerves of adult mice (Fig. 1b), we found that HDAC8 is exclusively localized in the cytoplasm of primary rat SCs cultured under conditions mimicking the conversion into the repair phenotype in normoxia or hypoxia (Fig. 4a). HDAC8 downregulation increased HIF1α levels under normoxic and hypoxic (Fig. 4b, c) conditions, and HIF1α downregulation by shRNA (Fig. 4d) reduced c-Jun expression and phosphorylation (Fig. 4e, f). We hypothesized that HDAC8 promotes HIF1α degradation. Indeed, when we inhibited the proteasome by a treatment with the proteasome inhibitor MG-132, HIF1α levels were no longer increased in cells where HDAC8 was downregulated compared to controls (Fig. 4g). Taken together, these data indicate that the increased levels of HIF1α in SCs mediated by HDAC8 downregulation are due to the protection of HIF1α from proteasomal degradation. Under normoxia, the Von Hippel-Lindau (VHL) tumor suppressor interacts with HIF1α and recruits a protein complex with E3 ubiquitin ligase activity that induces HIF1α ubiquitination and subsequent targeting to the proteasome[31]. However, expression levels of VHL were increased in normoxic conditions in SCs where HDAC8 was downregulated by shRNA (Supplementary Fig. 17), probably to

compensate the increased levels of HIF1α, as previously shown[48,49], suggesting that HDAC8 regulates HIF1α stability by a VHL-independent mechanism. Consistent with our cell culture findings, we show that HIF1α levels are increased in sciatic nerves of HDAC8 KO mice compared to control mice already at 1 dpl (Fig. 4h), indicating that HDAC8 KO SCs can rapidly upregulate HIF1α after injury. Interestingly, we identified this early increase of HIF1α levels in the cytoplasmic compartment, while nuclear levels of HIF1α were not affected (Fig. 4h), indicating that HIF1α-dependent regulation of c-Jun expression and phosphorylation at 1 dpl is mediated by a mechanism occurring in the cytoplasm. Similarly, at an early time point of hypoxia in rat SCs, the increase of HIF1α levels due to HDAC8 downregulation occurred in the cytoplasm, whereas no HIF1α was detected in the nucleus (Fig. 4i). Several studies have also reported high levels of HIF1α in the cytoplasmic compartment while nuclear levels remained low[50-52], suggesting a function of HIF1α in the cytoplasm. c-Jun is phosphorylated by JNK and in turn phosphorylated c-Jun dimerizes with ATF-2 to activate c-Jun expression[53]. In addition, the JNK pathway is known to mediate various hypoxia-induced cellular processes (reviewed in

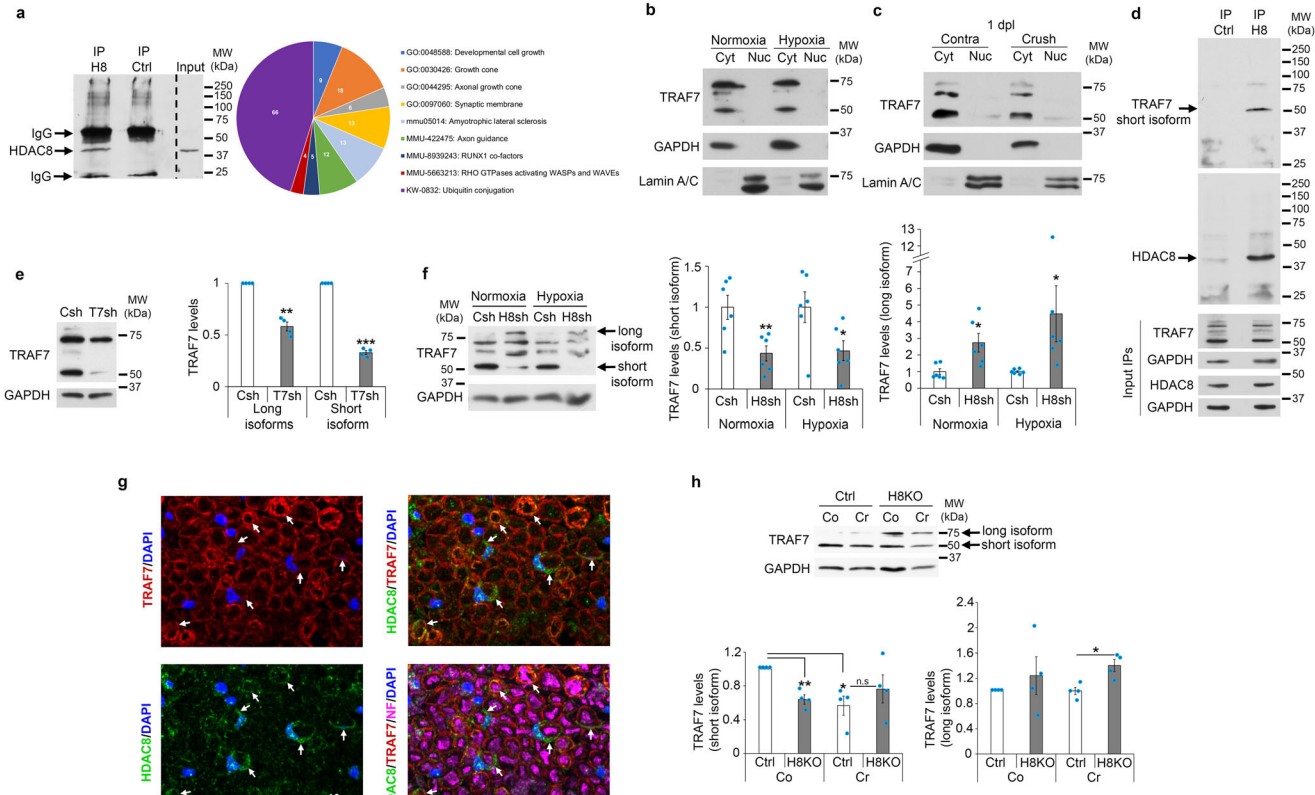

**Fig. 5 | HDAC8 interacts with and regulates TRAF7 expression pattern. a** HDAC8 (H8) putative binding partners GO analysis after H8 immunoprecipitation (IP) on lysates of crushed and contralateral mouse sciatic nerves at 1 dpl and analysis of binding partners by mass spectrometry. IP Ctrl = negative control IP on same lysates as H8 IP. **b, c** TRAF7 Western blot after subcellular fractionation (Cyt = cytoplasmic, Nuc = nuclear) of **b** rat SCs cultured under normoxia or hypoxia (CoCl$_2$ for 16 h) in conditions mimicking the conversion into the repair phenotype or **c** crushed (Crush) and contralateral (Contra) mouse sciatic nerves at 1 dpl. GAPDH and Lamin A/C = markers of cytoplasmic and nuclear fractions, respectively. **d** IP H8 or IP Ctrl and TRAF7 or HDAC8 Western blot in lysates of rat SCs cultured as above (**b**) in normoxia. Same lysate separated into two equal fractions (for IP H8 and IP Ctrl). TRAF7, HDAC8 and GAPDH inputs (3% of IP lysate) are shown. **e** TRAF7 and GAPDH Western blots in lysates of rat SCs cultured as above (**b**) in normoxia and incubated with lentiviruses carrying a TRAF7 shRNA (T7sh) or a control shRNA

(Csh), and quantification of short and long isoforms normalized to GAPDH. **f** TRAF7 Western blot in lysates of cells cultured as above (**b**) in normoxia and hypoxia and incubated with lentiviruses carrying an HDAC8 shRNA (H8sh) or Csh, and quantification normalized to GAPDH. **g** Co-immunofluorescence (z-series projections) of TRAF7 (red), HDAC8 (green) and Neurofilament (NF, magenta), and DAPI labeling (blue, nuclei) in cross-section of uninjured adult mouse sciatic nerves. **h** TRAF7 Western blot in lysates of HDAC8 KO (H8KO) and control (Ctrl) crushed (Cr) and contralateral (Co) mouse sciatic nerves at 1 dpl, and quantification normalized to GAPDH. Three independent IP (**a**), 3 animals (**c**), 5 independent experiments (**d**), sections of 3 WT mice analyzed (**g**), representative images are shown. $N$ = 3 (**b**), 4 (**e**) or 6 (**f**) independent experiments. **h** $N$ = 4 animals per group. Paired (**e**, **h**/Ctrl-Co) or unpaired (**f**, **h**/Ctrl-Cr) two-tailed (black asterisks) Student's t-tests, $p$ values: *<0.05, **<0.01, ***<0.001, n.s. = non-significant, values = mean, error bars = s.e.m. Source data are provided as a Source data file.

ref. 54). We found that in SCs cultured under hypoxic conditions, HIF1α downregulation by shRNA led to decreased activation of JNK (Fig. 4j), and JNK inhibition completely abrogated the increase of c-Jun phosphorylation and partly prevented c-Jun upregulation mediated by HDAC8 downregulation (Fig. 4k, l). Taken together, our data indicate that the ablation or downregulation of HDAC8 in SCs leads to HIF1α- and JNK-dependent c-Jun phosphorylation and upregulation.

### HDAC8 regulates TRAF7 to promote HIF1α degradation and prevent c-Jun phosphorylation

To understand how HDAC8 regulates HIF1α stabilization, we first asked whether HDAC8 deacetylase activity is involved in this function. We tested 2 different HDAC8 specific inhibitors, PCI-34051 and 1-Naphthohydroxamic acid, at different concentrations. None of the two HDAC8 inhibitors was able to increase HIF1α levels, c-Jun expression or phosphorylated c-Jun levels in SCs (Supplementary Fig. 18a–c), indicating that these effects are independent of HDAC8 deacetylase activity. HDAC8 has been previously shown to have in some instances deacetylase-independent functions, including acting as a scaffold for protein complexes or preventing the degradation of its binding partners[55–57]. To identify HDAC8 putative binding

partners, we immunoprecipitated HDAC8 in unlesioned mouse sciatic nerves and at 1 day post sciatic nerve crush lesion, and used a non-targeting IgG as immunoprecipitation control. We then analyzed co-immunoprecipitated proteins by mass spectrometry. We found 9 interesting, significantly enriched terms among which "Ubiquitin conjugation" showed the highest number of matching proteins (Fig. 5a and Supplementary Data 1). Among the putative binding partners of HDAC8 with a function in ubiquitin conjugation, we focused on tumor necrosis factor receptor-associated factor 7 (TRAF7) because it is localized in the cytoplasmic compartment in both primary rat SCs (Fig. 5b) and adult sciatic nerves (Fig. 5c) such as HDAC8, and possesses a RING domain with E3 ligase activity and can therefore mediate ubiquitination[58,59]. Our mass spectrometry analyses indicated TRAF7 as a putative binding partner of HDAC8, with 2 peptides identified for TRAF7 and 3 peptides identified for HDAC8, in both unlesioned nerves and nerves at 1 day post sciatic nerve crush lesion (Supplementary Data 1). We first confirmed that TRAF7 co-immunoprecipitated with HDAC8 in primary rat SCs (Fig. 5d). We used two different TRAF7 antibodies, which both detected three to four bands in primary rat SCs and in mouse sciatic nerves, two-three ranging from around 65 to 75 kDa and one shorter around 50 kDa

(Fig. 5b–d). Multiple isoforms of TRAF7 ranging from 65 to 75 kDa are commonly described. In addition, the NCBI database describes a shorter predicted isoform of 50 kDa (accession number: XR_007064922.1) that contains the RING finger domain with E3 ubiquitin ligase activity, the TRAF-type Zinc finger domain and the coiled-coil domain, but only one truncated WD40 repeat domain instead of the 7 WD40 repeat domains found in the longest TRAF7 isoform[59]. To test whether all bands detected by the TRAF7 anti-bodies are isoforms of TRAF7, we downregulated TRAF7 in primary rat SCs by using a lentivirus carrying a specific TRAF7 shRNA targeting a sequence common to all TRAF7 isoforms. Indeed, all four bands were downregulated (Fig. 5e), indicating that they correspond to four different isoforms of TRAF7. Among these four isoforms, we found that the shorter isoform co-immunoprecipitated with HDAC8 (Fig. 5d). Of note, the 2 TRAF7 peptides identified by mass spectrometry analyses are comprised within the short TRAF7 isoform (Supplementary Data 1). Interestingly, in primary rat SCs where HDAC8 was downregulated by shRNA, expression of the shorter TRAF7 isoform was significantly decreased, while expression of the longest isoform was increased (Fig. 5f). In sciatic nerves, we found that TRAF7 is expressed in all SCs (Fig. 5g), while as described above, HDAC8 is present only in sensory SCs and axons. Consistent with our cell culture findings, the short TRAF7 isoform was downregulated in HDAC8 KO nerves, while the long TRAF7 isoform was upregulated (Fig. 5h). Of note, the short TRAF7 isoform was also downregulated in crushed nerves of Control mice at 1 dpl compared to contralateral nerves, but not further downregulated in crushed nerves of HDAC8 KO mice compared to their contralateral nerves (Fig. 5h). This suggests that TRAF7 short isoform is completely downregulated already in SCs of contralateral nerves of HDAC8 KO mice and can therefore not be further downregulated. The remaining expression of TRAF7 short isoform is thus likely to be due to expression in non-sensory SCs. These results indicate that HDAC8 differentially regulates TRAF7 isoforms by stabilizing the short isoform and in contrast destabilizing the long isoform. We then asked whether TRAF7 is involved in HDAC8-dependent degradation of HIF1α and subsequent effects on c-Jun regulation. We found that TRAF7 downregulation by shRNA in primary rat SCs led to increased HIF1α levels in normoxic conditions (Fig. 6a) and increased c-Jun levels in hypoxic conditions (Fig. 6b), similar to the effects of HDAC8 downregulation. In addition, by co-immunoprecipitation, we show that the short TRAF7 isoform interacts with HIF1α in primary rat SCs (Fig. 6c, d) and that HIF1α ubiquitination is decreased in SCs where TRAF7 is downregulated by shRNA (Fig. 6e). However, in contrast to HDAC8 downregulation, we found that TRAF7 downregulation led to decreased levels of phosphorylated c-Jun in normoxia and has no effect on HIF1α and phospho-c-Jun levels in hypoxia (Supplementary Fig. 19a, b). It has been previously shown that TRAF7 can promote JNK phosphorylation[58,60,61], which in turn increases HIF1α levels and phosphorylates c-Jun[24,62–64]. Indeed, we show in primary rat SCs that TRAF7 downregulation leads to decreased phospho-JNK levels in normoxia but not in hypoxia (Supplementary Fig. 19c). It is highly likely that this effect is due to the downregulation of the long TRAF7 isoforms that contain WD40 repeats because TRAF7 is known to promote JNK phosphorylation by interacting with MEKK3 through its WD40 repeat domains[58,60]. As mentioned above, the short TRAF7 isoform contains only one truncated WD40 repeat domain and it was shown that a mutant of TRAF7 missing the WD40 repeat domains cannot interact with MEKK3 and that the long TRAF7 isoform cannot promote JNK phosphorylation in the absence of MEKK3[58]. Since HDAC8 stabilizes the short TRAF7 isoform but destabilizes the long TRAF7 isoform (Fig. 5f), we conclude that HDAC8 downregulation leads through the upregulation of the long TRAF7 isoform to increased phospho-JNK levels, inducing increased phospho-c-Jun levels and consequently increased c-Jun levels, and through the

downregulation of the short TRAF7 isoform to increased HIF1α levels, resulting in JNK-dependent increase of phospho-c-Jun and c-Jun levels and to JNK-independent increase of c-Jun levels. Consistently, we show here that overexpression of TRAF7 (long and short isoforms) reverted the increase of HIF1α and c-Jun but not of phospho-c-Jun levels mediated by HDAC8 downregulation through shRNA (Fig. 6f–i). Of note, all four TRAF7 isoforms were over-expressed by transfection of a construct carrying the longest iso-form, indicating that the isoforms detected by Western blot result either from post-translational modifications of the long isoform or potentially also from mRNA heterosplicing, a mechanism that has been recently described[65]. Taken together, these data show that HDAC8 oppositely regulates the short and the long TRAF7 isoforms, which leads to TRAF7-induced ubiquitination and degradation of HIF1α and slows down JNK phosphorylation, c-Jun phosphorylation and c-Jun upregulation in hypoxic conditions (Fig. 7).

## Discussion

While the functions of the class I HDACs HDAC1, HDAC2, and HDAC3 have been extensively studied in different areas of biology and medicine, the functions of HDAC8, the fourth and last member of the class I HDACs, have been comparatively less explored. It is, however, well established that HDAC8 mutations in humans affecting its deacetylase activity can cause subtypes of Cornelia de Lange syndrome, a genetic developmental disorder affecting multiple functions[66,67]. This syndrome is due to a loss of function of the cohesin complex and HDAC8 has been shown to deacetylate SMC3, a subunit of the cohesin complex, which is required for recycling the complex for subsequent cell divisions[68]. HDAC8 has also been involved in other pathological conditions including cancer, parasitic and viral infections, where it has mainly been shown to contribute to the pathology through its deacetylase activity[69–71]. In peripheral nerves, the functions of HDAC8 remained, however, unknown. We found that HDAC8 is upregulated after a sciatic nerve crush lesion in mice, suggesting a potential function in the degeneration and/or regeneration process after lesion. In this study, we thus set out to elucidate the functions of HDAC8 in SCs after a peripheral nerve injury.

Ablating HDAC8 specifically in adult SCs before a sciatic nerve crush lesion surprisingly did not impair regeneration, but instead promoted it, indicating a negative effect of HDAC8 on the regeneration process and demonstrating that this process is not optimal and can be improved. Interestingly, we found that HDAC8 ablation specifically promotes the regrowth of sensory axons and sensory function recovery. Indeed, we show that HDAC8 is exclusively expressed or expressed at high levels in SCs associated with sensory axons and that ablation of HDAC8 leads to increased levels of c-Jun and phosphorylated c-Jun in SCs. Parkinson et al. (2008)[72] have shown that both c-Jun and phospho-c-Jun are equally capable of inducing SC demyelination, and that overexpression of MKK7, which specifically phosphorylates JNK, leads to increased levels of c-Jun phosphorylation and of c-Jun, which induces c-Jun-dependent SC demyelination and conversion into repair SCs. Our findings strongly suggest that c-Jun- and phospho-c-Jun- mediated conversion into repair SCs is controlled by different mechanisms in sensory and motor SCs and identify HDAC8 as a marker of adult sensory SCs.

Mechanistically, we show here that HDAC8 counteracts hypoxia-induced c-Jun phosphorylation and upregulation in SCs by promoting the degradation of HIF1α and preventing the activation of JNK. Indeed, HIF1α and JNK are known to induce c-Jun phosphorylation and upregulation[20,21,46,53,54], and we show here that HIF1α downregulation in primary SCs cultured under hypoxic conditions leads to reduced JNK activation and that JNK inhibition prevents the increase of c-Jun phosphorylation and expression mediated by HDAC8 downregulation. HIF1α degradation in normoxic conditions can be mediated by VHL, which recruits a protein complex with E3 ubiquitin ligase activity that

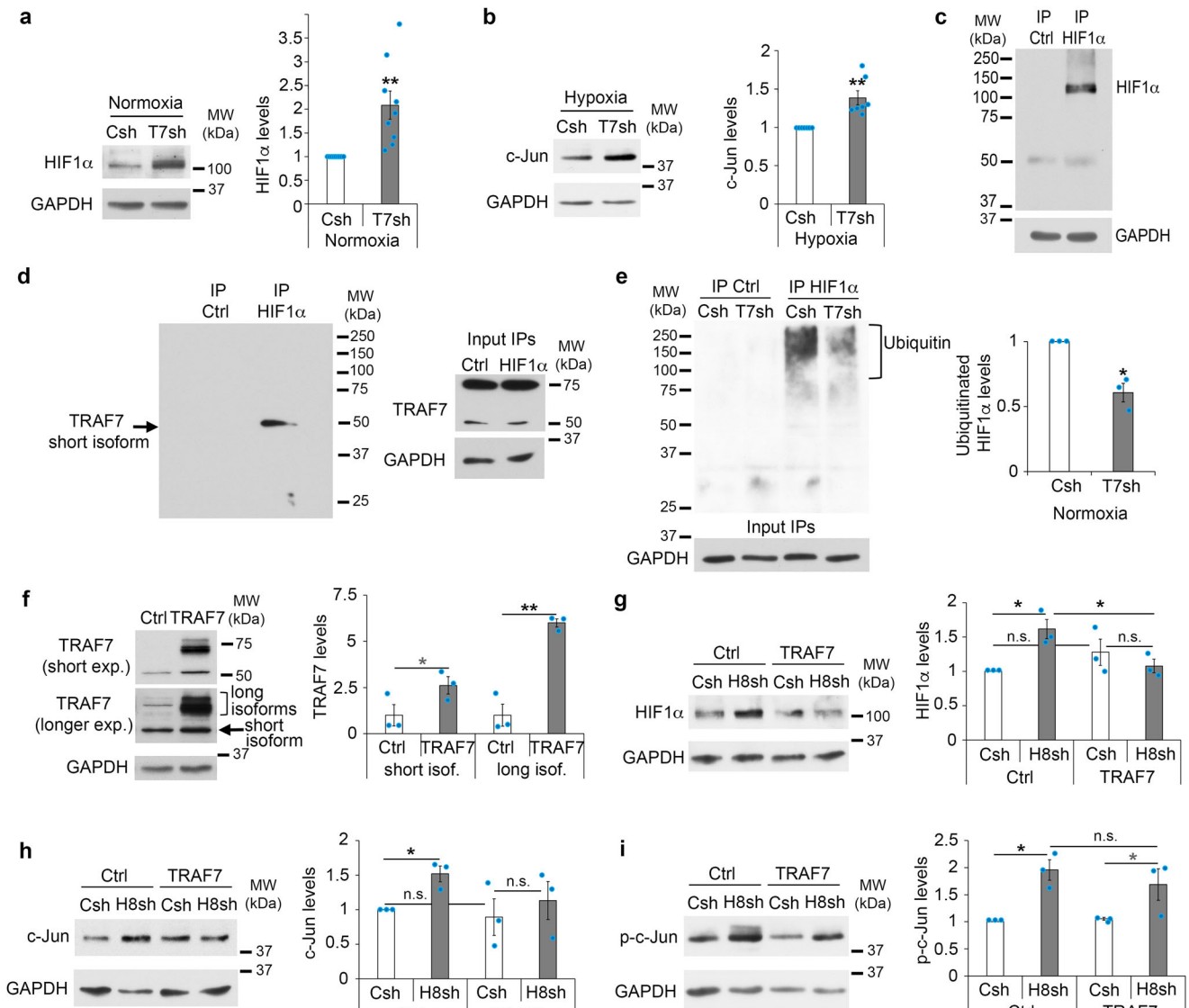

**Fig. 6 | HDAC8 controls HIF1α levels and JNK activation through TRAF7.**
**a**, **b** HIF1α (**a**) or c-Jun (**b**) Western blot in lysates of rat SCs cultured in conditions mimicking the conversion into the repair phenotype in normoxia (**a**) or hypoxia (**b**) and incubated with lentiviruses carrying a TRAF7 shRNA (T7sh) or control shRNA (Csh), and quantification normalized to GAPDH. $N = 9$ (**a**) or 7 (**b**) independent experiments. **c** HIF1α Immunoprecipitation (IP) or control IP (Ctrl) and HIF1α Western blot in lysates of rat SCs cultured as above (**a**) and incubated with MG-132 for 8 h. Same lysate separated into two equal fractions (for HIF1α IP and Ctrl IP). GAPDH input=3% of IP lysate. Three independent experiments, representative images are shown. **d** IP HIF1α or Ctrl IP and TRAF7 Western blot in lysates of rat SCs cultured as above (**a**) and incubated for 8 h with MG-132. Three independent experiments, representative images are shown. Same lysate separated into two equal fractions, (for HIF1α IP and Ctrl IP). TRAF7 and GAPDH inputs=3% of IP lysate. **e** Denaturing HIF1α and Ctrl IP and ubiquitin (P4D1) Western blot in lysates of SCs

transduced with T7sh or Csh lentivirus cultured as above (**a**) and incubated for 8 h with MG-132, and quantification of ubiquitinated HIF1α. $N = 3$ independent experiments. Each lysate separated into two equal fractions (for HIF1α IP and Ctrl IP). GAPDH input=3% of IP lysate. **f** TRAF7 Western blot on lysates of rat SCs transfected with a TRAF7- or Flag-expressing construct (Ctrl), and quantification normalized to GAPDH. $N = 3$ independent experiments. **g**–**i** HIF1α (**g**), c-Jun (**h**) or phospho-c-Jun (p-c-Jun, **i**) Western blots on lysates of rat SCs cultured as above (**b**) and incubated with lentiviruses carrying an HDAC8 shRNA (H8sh) or Csh, and subsequently transfected with a TRAF7- or Flag-expressing construct (Ctrl), and quantification normalized to GAPDH. $N = 3$ independent experiments. Paired (**a**, **b**, **e**, **g**, **h**, **i**/Ctrl) or unpaired (**f**, **i**/Csh-TRAF7) two-tailed (black asterisks) or one-tailed (gray asterisks, n.s.) Student's t-tests, $p$ value: *<0.05, **<0.01, n.s. = non-significant, values = mean, error bars = s.e.m. Source data are provided as a Source data file.

catalyzes HIF1α ubiquitination and thereby targets HIF1α to the proteasome for degradation[31]. VHL-independent regulation of HIF1α degradation has also been reported[73] and we show here that HDAC8-mediated HIF1α degradation in adult SCs is independent of VHL. By mass spectrometry analysis, we identified TRAF7 as a putative binding partner of HDAC8. We also showed that HDAC8 and TRAF7 are both exclusively or in large majority localized in the cytoplasmic compartment of SCs present in adult mouse sciatic nerves and in primary rat SCs in culture. By co-immunoprecipitation, we found that a short TRAF7 isoform, that had not been studied so far but is described as a

predicted isoform in the NCBI database, interacts with HDAC8 in SCs in vivo and in culture and mediates HDAC8-dependent degradation of HIF1α. TRAF7 possesses a RING finger domain with E3 ubiquitin ligase activity that is capable of catalyzing protein ubiquitination and thereby targeting proteins to the proteasome for degradation. The short 50-kDa TRAF7 isoform misses the C-terminal WD40 repeat domains found in the long TRAF7 isoforms. This structural difference confers to the short isoform different functions compared to the long TRAF7 isoforms regarding the activation of JNK. Indeed, TRAF7 is known to promote JNK phosphorylation and subsequent c-Jun phosphorylation

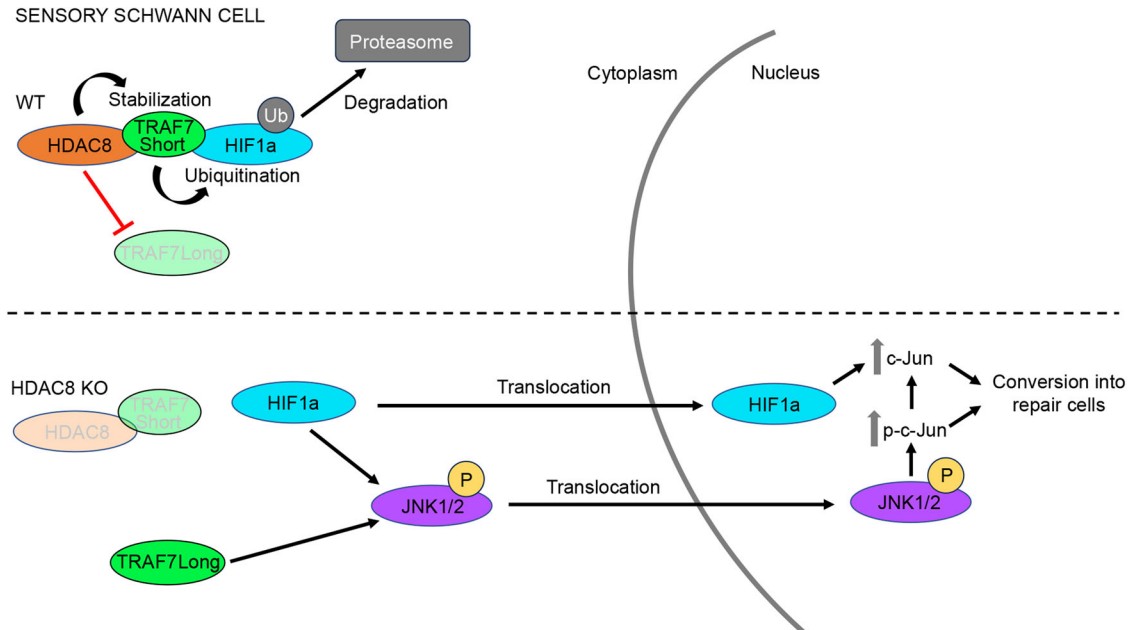

**Fig. 7 | Mechanism of action of HDAC8 in SCs after injury.** In WT SCs, HDAC8 interacts with and stabilizes TRAF7 short isoform, at the expense of TRAF7 long isoform; in turn, TRAF7 short isoform interacts with HIF1α and ubiquitinates it to target it to the proteasome for degradation. In the absence of HDAC8, TRAF7 short isoform is degraded, which leads to increased HIF1α levels. In parallel, TRAF7 long isoform is stabilized. Both HIF1α and TRAF7 long isoform increase the levels of phosphorylated JNK, HIF1α and phosphorylated JNK translocate to the nucleus, leading to increased phosphorylated c-Jun and total c-Jun levels. This mechanism promotes the conversion of sensory SCs into the repair phenotype and accelerates the regrowth of sensory axons and recovery of the sensory function.

by binding to MEKK3 through its WD40 repeat domains[60]. The short TRAF7 isoform missing these domains does not have this function. Instead, this isoform is stabilized, most likely through its interaction with HDAC8, and induces HIF1α ubiquitination and subsequent HIF1α degradation. Ablation of HDAC8 prevents this mechanism by leading to the downregulation of the short TRAF7 isoform and instead results in increased HIF1α levels and HIF1α-mediated c-Jun phosphorylation and upregulation. In addition, we show that ablation of HDAC8 leads to the upregulation of the long TRAF7 isoform, which potentiates JNK phosphorylation and thereby c-Jun phosphorylation and upregulation.

In summary, our study identifies a pathway controlled by HDAC8 that modulates HIF1α stabilization and JNK activation, and subsequent c-Jun phosphorylation and upregulation specifically in sensory SCs. This function of HDAC8 is not linked to its deacetylase activity, therefore HDAC8 inhibitors cannot be used to promote the regrowth of sensory axons and sensory function recovery. Instead, HDAC8 should be downregulated, either by using RNA interference or by repressing *Hdac8* gene activation. More work is needed to test whether such treatment strategies can be appropriately timed to promote regeneration. Alternatively, TRAF7 inhibitors or HIF prolyl hydroxylase inhibitors that enhance HIF1α stabilization[74] could also be tested to promote regeneration. Although our study did not identify a beneficial function of HDAC8-TRAF7 binding in SCs for PNS regeneration, we can speculate that by controlling HIF1α stabilization, the interaction of HDAC8 with TRAF7 may have beneficial effects in preventing neoangiogenesis and thereby the growth of tumors[30] such as schwannomas that are more susceptible to develop after a peripheral nerve injury[75].

## Methods
### Animals
Mice homozygous for floxed *Hdac8* allele (EMMA, Strain # EM:05717) were crossed with *PO*-CreERT2 or *Plp*-CreERT2 mice (mixed background back-crossed to *C57BL/6J*), in which the expression of the Cre recombinase (Cre) is tamoxifen-inducible and controlled by the Schwann cell-specific *PO* promoter or the Schwann cell and oligodendrocyte-specific *Plp* promoter[32], or with mice expressing the Cre recombinase under control of the *Dhh* promoter[42], leading to ablation of HDAC8 in SCs starting from embryonic day E13.5. Mice homozygous for floxed *Hif1α* allele (The Jackson Laboratory, Strain # 007561), were crossed with *Plp*-CreERT2 mice. To ablate HDAC8 or HIF1α in adult SCs, 3- to 4-month-old adult mice received daily injections of 2 mg tamoxifen (Sigma) for five consecutive days. As control mice, Cre-negative littermates of HDAC8 or HIF1α knockout mice were used. Genotypes were determined by PCR on genomic DNA.

Stratified random allocation in blocks with sex and age as strata was used. Analysis of the raw data was done at least twice by different researchers, independently. All mice used in this study were from mixed strains backcrossed to C57BL/6J. Mice were housed in a standard mouse facility with controlled temperature ($22 \pm 2\,°C$) and humidity ($45 \pm 10\%$) in individually ventilated cages containing sawdust bedding, a plastic cylinder and 2 paper tissues on the floor, food pellet and water ad libitum. Light cycle: 12:12 h. This study complies with all relevant ethical regulations concerning animal use, which was approved by the Veterinary office of the Canton of Fribourg, Switzerland, the Veterinary office (Landesuntersuchungsamt) of Rheinland-Pfalz, Germany, and the Veterinary office (Landesdirektion) in Sachsen, Germany.

### Surgical procedures
For all surgical procedures, isoflurane (3% for induction, 1.5–2% for narcosis during the operation, Isofluoran, CP Pharma) was used for anesthesia. For analgesia, 0.1 mg/kg body weight buprenorphine (Temgesic; Essex Chemie) was administered by i.p. injection 1 h before surgery and every four hours until the evening on the day of surgery. The following day, 1 ml agarose gel containing 0.027 mg/ml buprenorphine was given twice (morning and evening). The mice were placed on a heat pad during the entire procedure until waking up from anesthesia. To prevent dehydration of the eyes, a carbomer liquid eye gel (Lacryvisc, Alcon, or Viscotears, Novartis) was used preoperatively. Mice were shaved at the height of the hip for sciatic nerve crush lesion

and the field of operation was cleaned and disinfected. Sciatic nerve crush lesions were carried out on 3- to 4-month-old adult mice (males and females) as follows (by a procedure that we have previously described in ref. 12): An incision was made at the height of the hip and the sciatic nerve was exposed on one side. The nerve was crushed (5 times for 10 s with crush forceps: Ref. FST 00632-11). The wound was closed using Histoacryl Tissue Glue (BBraun). After the operation, mice were wrapped in paper towels until recovery from anesthesia.

## Functional recovery experiments

The motor function before lesion and its recovery were tested at 5, 12, 19 and 30 dpl using the Rotarod test and the Inverted grid test. For the Rotarod test, mice were placed four times on the Rotarod apparatus at a fixed speed of 15 revolutions per minute to test balance and motor coordination. The duration of each trial was limited to 1200 s. Latency to fall from the rotating beam was recorded and the average of the four trials was used for quantification. For the Inverted grid test, mice were placed on a cage grid, that was slowly inverted and observed for 30 s. The test was stopped if the animal fell. For each animal, the average count of grabs and missed steps as well as the average time the animals remained attached to the inverted grid among the four trials were calculated as described[76,77]. All trials were video-recorded for accurate analysis.

For both tests, mice were first trained three times and trials were separated by a 30 min recovery period.

The sensory function before lesion and its recovery were tested at 3, 5, 10, 12, 19, and 30 dpl by Toe pinch and von Frey tests. Toe pinch test: each toe of the rear foot on the right side (lesioned side) was pinched with equal pressure applied by the same experimenter using flat tip forceps. Immediate withdrawal was recorded as functional sensitivity of the pinched toe. In case no toe exhibited sensitivity, the same test was applied to toes of the contralateral side (uninjured side), which always resulted in immediate withdrawal. The von Frey test was conducted according to the Standard Operating Procedure of The Jackson Laboratory Mouse Neurobehavioral Phenotyping Facility: Mice were first acclimatized individually for 60 min in a small transparent cage with a grid floor. The test always begins with the left hind paw. As soon as the withdrawal threshold for the left hind paw is determined, the right hind paw is tested immediately afterwards. The filament (strength between 0.02 g and 8 g) is placed on the plantar surface of the foot with increasing pressure until it bends. First, the 0.4 g filament is applied to the plantar surface until it bends (~2–3 s). If no withdrawal reaction (snapping, licking the stimulated paw) is observed, the next higher filament in the row is applied and a possible withdrawal reaction is recorded again. Filaments with higher strength are tested one after the other on the mouse's paw until a withdrawal reaction is observed or the 8 g filament (maximum) is presented. As soon as a retraction reaction is observed, the test is repeated twice with the same filament. If a reaction is observed to the first 0.4 g filament, the test is continued with the filament of the next lower strength in the series until at least three reactions are absent or the lowest filament (0.02 g) is presented. The minimum threshold required to elicit paw withdrawal is averaged on each side, injured and uninjured, for each mouse. For the von Frey test, the same cohort of animals was tested at all time points, and we used a two-way mixed ANOVA test to evaluate if there is an interaction between genotype and time after lesion, explaining the mechanical force threshold, whereas genotype was used as between-subjects factor and time after lesion as within-subject factor. Post-hoc t-tests were used to identify significant differences between control and HDAC8 KO groups.

## Constructs

The c-Jun promoter-luciferase construct was generated by PCR on mouse genomic DNA: a 2.5 kb fragment corresponding to the mouse

c-Jun promoter (−1702/+781: 94939678-94942161 Mus musculus strain C57BL/6J chromosome 4, GRCm39) was inserted into the pGL3-Basic vector (Promega) between Mlu I and Bgl II restriction sites. The construct was verified by sequencing.

pCEP4F HDAC8-FLAG expression construct was a kind gift from Dr. Edward Seto. pLentiLox 3.7 and pLenti-C-mGFP were acquired from ATCC and Origene, respectively. pCXN2-Flag-TRAF7 and pCXN2-Flag were a kind gift from Dr. Huanjie Yang.

## Primary rat Schwann cell cultures

Purified primary rat SC cultures were derived from P2 Wistar rat sciatic nerves, as previously described in ref. 78. SCs were then purified by sequential immunopanning in plastic dishes coated with a Thy1.1 antibody[79]. Identity and purity were checked for each primary preparation by immunofluorescence of SC-specific markers (Sox10, Oct6, Krox20, P0, MAG). SCs were grown in proliferation medium: DMEM containing 10% FCS (Gibco), 1:500 penicillin/streptomycin (Invitrogen), 4 µg/ml crude GGF (bovine pituitary extract, Bioconcept), and 2 µM forskolin (Sigma), at 37 °C and 5% $CO_2$/95% air. SC culture protocol mimicking SC demyelination and conversion into repair cells that occur after a PNS lesion was as follows (previously described in ref. 12): SCs were first growth-arrested for 8 to 15 h in differentiation medium (DM): DMEM/F-12 (Gibco) containing 0.5% FBS (Gibco), 1:500 penicillin/streptomycin (Invitrogen), 100 µg/ml human apo-transferrin (Sigma), 60 ng/ml progesterone (Sigma), 1 µg/ml insulin (Sigma), 16 µg/ml putrescin (Sigma), 400 ng/ml l-thyroxin (Sigma), 160 ng/ml selenium (Sigma), 10 ng/ml triiodothyronine (Sigma) and 300 µg/ml bovine serum albumin (Sigma). Then 1 mM dbcAMP (Sigma) was added to induce differentiation and cells were incubated in this medium for another 3 days. The medium was then changed to DM only without dbcAMP, and RSCs were incubated in this medium for 3 days (differentiation mimicking adult SC stage). To mimic SC conversion into the repair phenotype, cells were then changed to proliferation medium and incubated in this medium for 1 day.

To inhibit HDAC8, 1 µM, 10 µM, or 100 µM PCI-34051 (HDAC8 inhibitor, Selleckchem), 30 µM or 90 µM 1-Naphthohydroxamic acid (HDAC8 inhibitor, MedChemExpress) or their vehicle were added in conditions mimicking the conversion into the repair phenotype at the time of change to proliferation medium.

To inhibit JNK, 1 µM JNK-IN-8 (JNK inhibitor, MedChemExpress) or its vehicle were added to rat SCs in conditions mimicking the conversion into the repair phenotype the day before and at the time of change to proliferation medium for 24 h.

To mimic hypoxia, 250 µM $CoCl_2$ was added to SCs in conditions mimicking the conversion into the repair phenotype 8 h after the change to proliferating medium, and incubated for 16 h. To induce hypoxia, RSCs were incubated in a hypoxic incubator (CO₂-Incubator C16, Labotect) in conditions mimicking the conversion into the repair phenotype, at the time of change to proliferation medium and for 16 h at 5%$CO_2$ and 1%$O_2$/94%$N_2$.

To inhibit the proteasome, 20 µM MG-132 (proteasome inhibitor, Hycultec) or its vehicle were added to rat SCs in conditions mimicking the conversion into the repair phenotype for 4 to 8 h.

HEK293T cells were obtained from ATCC and cultured in DMEM containing 10% FCS (Gibco) and 1:500 penicillin/streptomycin (Invitrogen) on poly-D-lysine (Sigma)-coated dishes.

Mycoplasma contamination was not tested in primary rat SCs or in HEK293T cells, because of the low incidence of mycoplasma contamination in primary cells, and because mycoplasma contamination results in inefficiency of transfection, which we did not observe in our primary rat SCs or in HEK293T cells.

## Generation and use of lentiviruses

To produce lentiviral particles, HEK293T cells (ATCC) were co-transfected with each lentiviral construct: HDAC8 shRNA

(TRCN0000087999), TRAF7 shRNA (TRCN0000302734), HIF1a shRNA (TRCN0000232221) or control shRNA (SHC001), together with the packaging constructs pLP1, pLP2 and pLP/VSVG (Invitrogen), using Lipofectamine 2000 (Invitrogen), according to the recommendations of the manufacturer (ViraPower Lentiviral Expression Systems Manual). Lentiviruses were added to primary rat SCs in DM and incubated for two days. Then, infected cells were selected by adding 2 µg/ml puromycin (Sigma) in DM for two days, before adding proliferating medium. HDAC8 shRNA, TRAF7 shRNA, HIF1-α shRNA and control shRNA lentiviral constructs were purchased from Sigma-Aldrich.

### Transfection

Confluent primary rat SCs were transfected either in differentiating conditions or in conditions mimicking the conversion into the repair phenotype at the time of change to proliferation medium with Fugene 6 (Promega) at 5:1 ratio Fugene 6:DNA, according to the manufacturer's recommendations, with the following modifications: the DNA was incubated in OptiMEM (Gibco) for 5 min at room temperature (RT) before addition of Fugene 6. The mix Fugene 6:DNA was then incubated for 30 min at RT before being added to the cells.

### Luciferase gene reporter assay

Confluent primary rat SCs previously infected in differentiating conditions with HDAC8 shRNA or control shRNA in 12-well plates were transfected in conditions mimicking the conversion into the repair phenotype at the time of change to proliferation medium with Fugene 6 (Promega) at 3:1 ratio Fugene 6:DNA, according to the manufacturer's recommendations, with the following modifications: the DNA was incubated in OptiMEM (Gibco) for 5 min at RT before addition of Fugene 6. The mix Fugene 6:DNA was then incubated for 30 min at RT before being added to the cells. Eight hours after transfection, 250 µM $CoCl_2$ or vehicle ($H_2O$) were added to the cells for 16 h. Cells were lysed 24 h after transfection in 110 µl Reporter Lysis Buffer (Promega) and assayed for luciferase activity. Forty microliters of lysate were subjected to two consecutive injections of each 25 µl luciferase substrate and values were recorded after an integration time of 2 and 10 s, respectively. Efficiency of transfection was evaluated by measuring beta-galactosidase activity of a co-transfected beta-gal construct (Promega). Primary rat SCs were co-transfected with 1 µg of the luciferase construct pGL3b-c-Jun promoter or empty pGL3b (Promega), and 300 ng beta-gal construct (Promega). Luciferase activity was first normalized to beta-galactosidase activity and then normalized to the empty pGL3b luciferase control (transfected at the same ratio Fugene 6:DNA). For beta-galactosidase activity, we followed the instructions of the manufacturer (Promega). Briefly, 50 µl Assay Buffer 2x was added to 50 µl lysate, and beta-galactosidase activity was recorded after a 15–20 min incubation time at 37 °C. Endogenous beta-galactosidase activity (as determined from cells only transfected with overexpressing construct at the same ratio Fugene 6:DNA) was subtracted.

### RT-PCR

Isolation of RNA was carried out using Trizol reagent (Invitrogen) and cDNA was produced using SuperScript II Reverse Transcriptase (Thermo Scientific), according to the manufacturers' recommendations. Quantitative real-time PCR analyses were performed with an ABI 7000 Sequence Detection System (Applied Biosystems) using FastStart SYBR Green Master (Roche), according to the manufacturer's recommendations. A dissociation step was added to verify the specificity of the products formed. Primer sequences were as follows:

for rat *c-Jun*, forward 5′-CACCTCCGAGCCAAGAACTC-3′, reverse 5′-CTGGACTGGATGATCAGGCG-3′

for rat *Gapdh*, forward 5′-GTATCCGTTGTGGATCTGACAT-3′, reverse 5′-GCCTGCTTCACCACCTTCTTGA-3′

### Western blot analysis

For all Western blots and immunoprecipitations on mouse sciatic nerve lysates, we collected the injured sciatic nerve from the lesion site to around 12 mm distal to the lesion site (region where the nerve splits into the three branches of tibial, sural and common peroneal nerves). We collected the same region of the contralateral nerve as internal control for each animal. After perineurium removal, sciatic nerves were frozen in liquid nitrogen, pulverized with a chilled mortar and pestle, lysed in radioimmunoprecipitation assay (RIPA) buffer (10 mM Tris/HCl, pH 7.4, 150 mM NaCl, 50 mM NaF, 1 mM NaVO4, 1 mM EDTA, 0.5% w/v sodium deoxycholate, and 0.5% Nonidet P-40) for 30 min on ice, and centrifuged to pellet debris. Supernatants were collected.

Cells were washed once in PBS, lysed in RIPA buffer for 15 min on ice, and centrifuged to pellet debris. Sciatic nerves and cell lysates were submitted to SDS-PAGE and analyzed by Western blotting.

Primary antibodies used: HDAC8 (rabbit, 1:1000, Santa Cruz Biotechnology, cat. # sc-11405, lot # D0715), HDAC1 (rabbit, 1:1000, GeneTex, cat. # GTX100513, lot # 39471), HDAC2 (mouse, 1:1000, Sigma, cat. # H2663, lot # 092M4824V), HDAC3 (rabbit, 1:1000, GeneTex, cat. # GTX109679, lot # 39974), Oct6 (rabbit, 1:2000, kind gift from Dr. Dies Meijer, University of Edinburgh), Krox20 (rabbit, 1:500, kind gift from Dr. Dies Meijer, University of Edinburgh), Sox10 (mouse, 1:1000, Abcam, cat. # ab216020, lot #GR3272630-2), P0 (chicken, 1:5000, Aves Labs, cat. # PZO, lot # PZO877982), MBP (rat, 1:759, Bio-Rad, cat. # MCA409S, lot # 158446), c-Jun (rabbit, 1:1000, Cell Signaling, cat. #9165, lot # 13), phospho-c-Jun (rabbit, 1:500, Cell Signaling, cat. #3270, lot # 5), TRAF7 (rabbit, 1:1000, Proteintech, cat. # 11780-1-AP, lot # 00047291), TRAF7 (rabbit, 1:1000, ABclonal, cat. # A3095, lot # 5500008204), HIF1α (rabbit, 1:1000, Novus Biological, cat. # NB100-479, lot # D108267-1), HIF1α (mouse, 1:500, R&D Systems, cat. # MAB1536, lot # KRK0521051), VHL (rabbit, 1:1000, ABclonal, cat. # A0377, lot # 5500020894), Ubiquitin (P4D1) (mouse, 1:1000, Cell Signaling, cat. # 3936, lot # 19), phospho-JNK1/JNK2 (rabbit, 1:1000, Invitrogen, cat. # 44-682 G, lot # 2465206), GAPDH (mouse, 1:5000, GeneTex, cat. # 28245, lot # 822203823), ß-Actin (mouse, 1:5000, Sigma, cat. # A5441, lot # 122M4782), Lamin A/C (mouse, 1:1000, Sigma, cat. # SAB4200236, lot # 055M4822V), eEF1A1 (rabbit, 1:5000, Abcam, cat. # ab157455, lot #GR231741-10).

All secondary antibodies were from Jackson ImmunoResearch: light-chain specific goat anti-mouse-HRP (horse radish peroxidase) and goat anti-rabbit-HRP, heavy-chain-specific goat anti-chicken-HRP.

### Immunoprecipitation

For non-denaturing IPs, tissues or cells were processed as described in the Western blot section in the following RIPA buffer: 10 mM Tris/HCl, pH 7.4, 150 mM NaCl, 50 mM NaF, 1 mM NaVO4, 1 mM EDTA, 0.5% w/v sodium deoxycholate, 0.5% Nonidet P-40 and protease inhibitor cocktail (Thermo Scientific, cat. #78443). Lysates were pre-cleared for 1 h at 4 °C with 30 µl protein A/G PLUS agarose beads (Santa-Cruz Biotechnology). Antibodies were conjugated to 30 µl of agarose beads for 2 h at 4 °C on a rotating platform. Then, 1 mL of cleared lysate was rotated for 2 h at 4 °C with immunoprecipitating antibodies conjugated to the beads.

For denaturing IPs, tissues or cells were lysed in 10 mM Tris/HCl, pH 7.4, 1% SDS, boiled for 5 min at 95 °C, and mixed with 9 volumes of RIPA buffer without SDS. Lysates were homogenized with a 20-gauge needle attached to a sterile plastic syringe, pre-cleared for 1 h at 4 °C with 30 µl of agarose beads, and rotated overnight at 4 °C with immunoprecipitating antibodies. The day after, 30 µl of agarose beads were added and samples were further rotated at 4 °C for 2 h.

Two to six micrograms of the following antibodies were used per nerve or per $1 \times 10^7$ cells: HIF1α (mouse, R&D Systems, cat. # MAB1536, lot # KRK0522111), HIF1α (rabbit, Novus Biological, cat. # NB100-479, lot # D108267-1), HDAC8 (sheep, R&D Systems, cat. # AF4359, lot # CAKS0120091), Normal Goat IgG control (goat, R&D Systems, cat. #

AB-108-C, lot # ES41160812), Flag (mouse, Sigma, cat. # F1804, lot # SLBM0089V), GFP (rabbit, Abcam, cat. # ab290, lot # GR3431263-1). Rabbit anti-GFP, mouse anti-Flag and goat anti-IgG antibodies were used as control IPs.

Immunoprecipitates were pelleted, washed four times with RIPA buffer, eluted in 40 μl of Laemmli buffer and incubated at 75 °C for 10 min. Analysis was done by Western blot.

## Subcellular fractionation

Cytoplasmic and nuclear fractions were obtained by following the protocol described in ref. 80. For fractionation on sciatic nerve lysates, we collected the injured sciatic nerve from the lesion site to around 12 mm distal to the lesion site (region where the nerve splits into the three branches of tibial, sural, and common peroneal nerves). We collected the same region of the contralateral nerve as internal control for each animal. After perineurium removal, sciatic nerves were pulverized with a chilled mortar and pestle and lysed in hypotonic buffer (20 mM Tris/HCl, pH 7.4, 10 mM KCl, 2 mM $MgCl_2$, 1 mM EGTA, 0.5 mM DTT, and 0.5 mM PMSF) for 30 min on ice.

Cells were harvested by trypsination, resuspended in PBS, centrifuged at $500 \times g$ for 3 min at 4 °C and resuspended in hypotonic buffer for 3 min on ice. Then, Nonidet P-40 was added to a final concentration of 0.1% and the lysates were further incubated for 3 min and centrifuged at $1000 \times g$ for 3 min at 4 °C. The supernatant (cytoplasmic fraction) was centrifuged a second time at $15,000 \times g$ for 3 min at 4 °C to pellet remaining debris. The pellet (nuclear fraction), was resuspended in isotonic buffer (20 mM Tris/HCl, pH 7.4, 150 mM KCl, 2 mM $MgCl_2$, 1 mM EDTA, 0.3% Nonidet P-40, 0.5 mM DTT, and 0.5 mM PMSF) on ice for 10 min, and centrifuged at $1000 \times g$ for 3 min at 4 °C. The pellet was then resuspended in RIPA buffer (10 mM Tris/HCl, pH 7.4, 150 mM NaCl, 50 mM NaF, 1 mM $NaVO_4$, 1 mM EDTA, 1% w/v sodium deoxycholate, 1% Nonidet P-40, 1% SDS) with protease inhibitors (Thermo Scientific, cat. # 78443) for 30 min and centrifuged at $2000 \times g$ for 3 min at 4 °C. Supernatants were collected.

## Immunofluorescence and imaging

Mouse sciatic nerves were fixed in situ with 4% PFA for 10 min, dissected, embedded in PolyFreeze (Sigma), and frozen at −80 °C. In some cases, the sural, peroneal and tibial nerves were collected separately. Nerve cryosections (5-μm thick) were first incubated with acetone for 10 min at −20 °C, washed in PBS, blocked for 1 h at RT in blocking buffer (0.3% Triton X-100/10% BSA/PBS), and incubated with primary antibodies and in some cases also with Isolectin GS-B4, Alexa Fluor™ 594 conjugate (Invitrogen) at a 1:200 dilution overnight at 4 °C in blocking buffer without Triton X-100. Sections were then washed 3 times in blocking buffer and secondary antibodies were incubated for 1 h at RT in the dark in blocking buffer without Triton X-100. Sections were washed again, incubated with DAPI for 5 min at RT, washed and mounted in Citifluor (Agar Scientific).

Hind paws and ventral roots were collected after mouse perfusion with 4% PFA. Ventral roots and hind paws were post-fixed in 4% PFA for 1 h at RT or overnight at 4 °C, respectively, washed 3 times with PBS, and cryoprotected by incubation in 30% sucrose at 4 °C for 24 h. Palmar skin of each hind paw was dissected out before incubation in sucrose. Tissues were then embedded in PolyFreeze (Sigma) and frozen at −80 °C. Cryosections (16-μm thick for hind paw skin samples and 5-μm thick for ventral roots samples) were dried for 5 h at RT, and then submitted to antigen retrieval in pre-heated sodium citrate buffer with 0.05% Tween 20 at 80 °C for 20 min. Sections were then washed 3 times in PBS and incubated in blocking buffer (2% BSA/0.3% Triton X-100 in PBS) for 1 h before incubation with primary antibodies overnight at 4 °C in blocking buffer without Triton X-100. Sections were then washed 3 times in blocking buffer and incubated with secondary antibodies in blocking buffer without Triton X-100 for 1 h at RT for ventral root sections and for 2 h for hind paw skin sections. Sections were then washed in blocking buffer with Triton X-100, incubated with DAPI for 5 min, washed with PBS, and mounted in Citifluor (Agar Scientific).

For whole-mount sciatic nerve staining, nerves were processed as described[81] with the following modifications: the perineurium was fully removed after 5 h fixation in 4% PFA and each nerve branch was separated from one another.

Primary antibodies: HDAC8 (sheep, 1:200, R&D Systems, cat. # AF4359, lot # CAKS0120091), TRAF7 (rabbit, 1:200, ABclonal, cat. # A3095, lot # 5500008204), c-Jun (rabbit, 1:200, Cell Signaling, cat. #9165, lot # 13), phospho-c-Jun (rabbit, 1:800, Cell Signaling, cat. #3270, lot # 5), F4/80 (rat, 1:200, GeneTex, cat. # GTX26640, lot # 821401311), NF-M (chicken, 1:1000, GeneTex, cat. # GTX85461, lot # 822102296), NF-M (rabbit, 1:200, Merck Millipore, ref. AB1987, lot # 3956946), Stathmin-2 (rabbit, 1:1000, Novus Biologicals, cat. # NBP1-49461, lot # D-1), Stathmin-2 (goat, 1:200, Origene, ref. TA318234, lot # 22900), GAP43 (rabbit, 1:500, Abcam, cat. # ab75810, lot #GR3299539-7), P0 (chicken, 1:1000, Aves lab, ref. PZO, lot # 877982). All secondary antibodies were from Jackson ImmunoResearch. Photos were acquired using a Visitron VisiScope spinning disk confocal system CSU-W1 with the VisiView® 6.0 software. Fiji (version 1.0) and Adobe Photoshop (CC 25.6.0 Release) were used to process images. z-series projections (stated in the figure legends) are shown. Axonal regrowth was measured using NeuronJ software (ImageJ plugin freely available online with a user manual: http://www.imagescience.org/meijering/software/neuronj/). Parameters used were the same as described in the method validation article[82]: Neurite appearance: Bright, Hessian smoothing scale: 2.0, Cost weight factor: 0.7, Snap window size: 9 × 9, Path-search window size: 2500 × 2500, Tracing smoothing range: 5, Tracing sub-sampling factor: 5, Line width: 1.

## Electron microscopy

Mice were killed with 150 mg/kg pentobarbital i.p. (Esconarkon; Streuli Pharma AG) and sciatic nerves were fixed in situ with 4% paraformaldehyde (PFA) and 3% glutaraldehyde in 0.1 M phosphate buffer, pH 7.4. Fixed tissues were post-fixed in 2% osmium tetroxide, dehydrated through a graded acetone series as described[83] and embedded in Spurr's resin (Electron Microscopy Sciences). Semithin sections were stained with 1% Toluidine blue for analysis at the light microscope, and ultrathin sections (70-nm thick) were cut as described[83]. All analyses were done in the sciatic nerve at 5 mm distal to the lesion site. No contrasting reagent was applied. Images were acquired using a Philips CM 100 BIOTWIN equipped with a Morada side mounted digital camera (Olympus).

## Mass spectrometry analysis

Protein samples eluted from the beads in 1X Laemmli sample buffer were heated for 10 min at 75 °C, treated with 1 mM DTT, and alkylated using 5.5 mM iodoacetamide for 10 min at RT. Samples were fractionated on 4–12% gradients gels and proteins were in-gel digested with trypsin (Promega) into four fractions per sample. Tryptic peptides were purified by STAGE tips and LC-MS/MS measurements were performed on a QExactive HF-X mass spectrometer coupled to an EasyLC 1200 nanoflow-HPLC (all Thermo Scientific) as described[84]. MaxQuant software (version 1.6.2.10)[85] was used for analyzing the MS raw files for peak detection, peptide quantification and identification using a full length Uniprot mouse database (version April 2016). Carbamidomethylcysteine was set as fixed modification and oxidation of methionine was set as variable modification. The MS/MS tolerance was set to 20 ppm and four missed cleavages were allowed for Trypsin/P as enzyme specificity. Based on a forward-reverse database, protein and peptide FDRs were set to 0.01, minimum peptide length was set to seven, and at least one unique peptide per protein had to be identified. The match-between run option was set to 0.7 min. MaxQuant results were analyzed using Perseus software (version 1.6.2.10)[86].

## Statistical analyses

For each dataset presented, experiments were performed at least three times independently or with at least three animals and $p$ values were calculated in Microsoft Excel (Mac version 16.91) using two-tailed (black asterisks) or one-tailed (gray asterisks) Student's t-tests. $P$ values: *<0.05, **<0.01, ***<0.001, data are presented as mean values ± SEM. For data sets obtained using animals or their tissues, data and tissues of each animal were processed independently. Sample size was determined by the minimal number of animals or individual experiment required to obtain statistically significant results and increased in some cases to improve confidence in the results obtained, $n = 3$ to 12. Sample size was calculated for a power of 0.8. No animal or data point was excluded from the analysis.

## Reporting summary

Further information on research design is available in the Nature Portfolio Reporting Summary linked to this article.

## Data availability

The mass spectrometry proteomics data have been deposited to the ProteomeXchange Consortium via the PRIDE [1] partner repository with the dataset identifier PXD046582. Source data including uncropped blots of main Figures and raw data of all figures are provided with this paper. Uncropped blots of Supplementary Figs. are presented at the end of the Supplementary Information file. Source data are provided with this paper.

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

## Acknowledgements

*Plp*-CreERT2 and *P0*-CreERT2 mice have been used in collaboration with Dr. Ueli Suter (ETH Zürich, Switzerland). We thank Dr. Huanjie Yang (School of Life Science and Technology, China) and Dr. Edward Seto (GW Cancer Center, USA) for the TRAF7- and HDAC8-expressing constructs, respectively, Dr. Jorge Pereira (ETH Zürich, switzerland) for technical advice, Dr. Elmo Neuberger (JGU Mainz, Germany) for support with statistical analyses, and Dr. Regnier-Vigouroux (JGU Mainz, Germany) for access to the hypoxia incubator. Swiss National Science Foundation grants PP00P3_1139163 (C.J.), PP00P3_163759 (C.J.) and 31003A_173072 (C.J.), Deutsche Forschungsgemeinschaft grant JA 3019/6-1 (C.J.).

## Author contributions

N.H. and C.J. conceived and designed the experiments. N.H., M.D., M.B., V.B., M.G., F.S., F.D., R.F., R.S., D.M., D.S.S., and S.R.L. generated reagents and performed the experiments. N.H., M.B., D.S.S., J.D., and C.J. analyzed the data, N.H. and C.J. wrote the manuscript. All authors commented on the manuscript.

## Funding

## Competing interests

Patent application # EP24191599, filed in Europe on 29 July 2024, title: Compounds and Compositions for Treating PNS Injury and Degenerative Diseases. The authors declare no other competing interests.
