## [Peer review file · Nature Communications]

Hypoxia-induced conversion of sensory Schwann cells into repair cells is regulated by HDAC8

Corresponding Author: Professor Claire Jacob

Version 0:

Reviewer comments:

Reviewer #1

(Remarks to the Author)

This body of work has been performed by the Jacob lab who are leaders in the roles of HDACs in the nervous systems and the molecular regulation of the Schwann cell phenotype during nerve injury. In their current work they demonstrate that HDAC 8 in Schwann cells partly inhibits JUN upregulation, a central regulator of nerve injury, and this is achieved through TRAF7 and HIF1alpha. What is very intriguing about the study is that Schwann cell HDAC8 appears to be a break for sensory axon regeneration. The authors postulate that this may identify differential regulation of the Schwann cell repair phenotype in sensory versus motor associated Schwann cells. While the data presented in this manuscript are generally solid the experiments don't go quite far enough to support some of the authors claims in the abstract and the discussion. I do believe with the below additions this paper would be a substantial contribution to the literature.

I suggest the following amendments:

- 1) In the abstract and line 35/36 and Line 324 – A central claim of this paper is that the conversion into repair cells is controlled by different mechanisms in sensory and motor Schwann cells and the findings in the HDAC8 cKO identify this. The authors never actually look at whether motor Schwann cells phenotype is affected in the HDAC8 cKO. I would suggest the authors need to substantiate this claim by quantifying % Jun +/- Jun +ve Schwann cells in sensory vs motor Schwann cells at 1 day post injury in HDAC8 KO nerves compared to Wt. Otherwise, the authors need to substantially soften this conclusion in the abstract and the discussion.
- 2) Fig1F and Fig S4. Another central findings is the HDAC8 positive Schwann cells only surround sensory axons, however the authors never quantify this. Can the authors quantify the number of HDAC8 +ve SCs surrounding stathmin 2 +ve and -ve axons.
- 3 a) Fig.S1a – shows 1/3rd reduction in HDAC8 levels by WB in PlpCreERT2 HDAC8 KO and Fig.S1b – shows a 50% reduction in HDAC8 levels by WB I MPZCreERT2 HDAC8 KO. Of course, some of the remaining HDAC8 may be in axons. Presumably axonal HDAC8 is unaffected in the mutants?
- 3 b) Related to 3 a) The PlpCreERT2 tends to flox out more in Remak Schwann cells than myelin ones whereas the MPZCreERT2 wont flow out in any Remak/unmyelinated Schwann cells. I think the authors need to make it clear to the reader what % of recombination there is in myelinated and unmyelinated and motor versus sensory (ideally sensory myelinated and sensory unmyelinated by I appreciate this may be difficult to do).
- 4) Related to the above point the authors don't discuss the different types of sensory axons and Schwann cells. There are A alpha and A beta large myelinated sensory fibres, A delta thinly myelinated sensory fibres and unmyelinated C fibres. Is HDAC8 involved in all or some of these? If the authors are unable to provide data to investigate this then I would accept discussion of this point
- 5) Where is the uninjured analysis of Control and HDAC8 cKO for both PlpCreERT2 and MPZCreERT2– axon counts, g-ratio, SC numbers? Please add this to supplemental.
- 6) Fig. 1C Would be better to quantify demyelination at one or two earlier time points (say day 3 and 7) as 12 days is very

late.

7) Fig. 1 Would be better to display functional recovery data in line graphs with data points connected by lines over time. It is odd that the time points start at day 12 post injury. Better to plot pre-lesion tests (was there a difference between control and KO uninjured?) and then there should be a few earlier time points prior to visualising regeneration in WT or mutants – Firstly to show that nerve crushes were complete between genotypes and biological replicates and secondly to demonstrate that there is precocious axon regeneration.

8) Fig. 1d. The display of the data for the toe pinch is odd as all the toes are grouped together in the display yet the time to regeneration of each toe is slightly different – digit 1 being the fastest (day 13) and 5 the slowest (day 16) – see Siconolfi and Seeds 2001, DOI: <https://doi.org/10.1523/JNEUROSCI.21-12-04348.2001>. It would be better to display the data for each individual toe as I've outlined in point 3. Some of this data could be put in supplemental.

9) Fig 1d. I would suggest an additional method to test functional sensory recovery between control and HDAC8 and cKO such as Von Frey hair and/or Hargreaves test if available.

10) Fig S7 the down regulation of c-Jun in Hif1alpha flox nerves at 3 days is fairly slight. There are only 3 data points for the mutant – 2 of which are equivalent to the control and one animal with 50% the JUN level. Given that this reported finding is a key conclusion that the proposed mechanism hinges on I would suggest the authors increase the number of mutants to 5 and do additional time points after injury say 1 or 2 days as well as 5 or 7 days post injury. Also, this data should be in the main figure panels not in supplemental.

11) Related to Fig S7 What is the phenotype of the Hif1alpha cKO mouse – does it have slower myelin clearance/regeneration? As Hif1alpha is proposed as a key regulator of JUN it is important to show this data.

12) Discussion + Fig5. A lot of weight is put on the role of JNK and JUN phosphorylation in the authors proposed mechanism. The authors need to discuss the evidence for JNK and Jun phosphorylation in nerve injury. My understanding is N terminal phosphorylation is not required based on Parkinson 2008 doi/ JCB 62510.1083/jcb.200803013 and data supporting a role for JNK in nerve injury is solely from in vitro studies.

Additional points:

A) Introduction: In addition to Shh add in mention of additional 2 known regulatory mechanisms of c-JUN in Schwann cells after injury - mTOR regulates c-JUN protein levels – Normén et al., 2018 DOI:10.1523/JNEUROSCI.3619-17.2018 and Kim et al., 2018 O-GlcNAcylation of JUN and inhibits transcription activity allowing remyelination doi/10.1073/pnas.1805538115

B) Line 119 – These are not Remak Schwann cells/axons but regeneration clusters

C) Figure 1 C – For EM analysis – how many sections per nerve were quantified and which fascicle of the sciatic was quantified (peroneal/tibial/sural) or where all 3 quantified.

D) Fig 1F – Is this a maximum projection or individual z position?

E) Fig 3i the WB for p-JNK does not seem representative of the quantifiable difference (a 25% reduction)

F) Discussion: Authors should discuss the findings of Babetto et al., 2020. They floxed HIF1alpha out of Schwann cells in nerve injury and showed less phosphorylated S6 kinase and slightly increased axon disintegration after injury at 36hours.

G) In general authors should try to use colourblind friendly colour combinations for immunos – avoid red/green

Reviewer #2

(Remarks to the Author)

Hertzog et al. investigated the role of HDAC8 in the conversion of sensory Schwann cells into repair cells under hypoxic conditions. They first show that HDAC8 expression is upregulated in peripheral nerves after lesion. They then show that ablation of HDAC8 in adult Schwann cells results in less myelin debris and more axons per surface area at 12 dpl. Using primary rat SCs cultured under conditions that promote conversion of SCs into the repair phenotype, they show that downregulation of HDAC8 led to increased levels of c-Jun and phosphorylated c-Jun. These effects were more pronounced when cells were incubated under hypoxic conditions. Therefore, the remaining studies were performed in hypoxic conditions. The authors showed that HDAC8 downregulation increased HIF1a levels in both normoxic and hypoxic conditions. They hypothesized that HDAC8 regulates HIF1a stability via a VHL-independent mechanism. They also suggest that downregulation of HDAC8 in SCs leads to HIF1a- and JNK-dependent c-Jun phosphorylation and upregulation. Lastly, they investigated the potential binding partners of HDAC8 by mass spectrometry. They identified TRAF7 as a putative binding partner HDAC8. They conclude that HDAC8 regulates the short and long TRAF7 isoforms, leading to ubiquitination and degradation of HIF1a and inhibition of the JNK pathway. The results suggest an interesting mechanism for conversion of Schwann cells into repair cells under hypoxic conditions. However, there are some major concerns regarding the lack of controls in the experiments, and conclusions stated by the authors that are not sufficiently substantiated by the data presented. Furthermore, the figures are difficult to understand without schematics for each experimental design, and the

manuscript would benefit from editing to improve flow and clarity. This manuscript needs significant experimental and editorial revisions to be suitable for publication in Nature Communications.

Major Comments:

1. The authors state that their HDAC8 KO mouse line is specific to Schwann cells. However, PLP and P0 are expressed in satellite glial cells. Elbaz et al 2022 (PMC9354470) used the PLP-CreERT-ROSA26-Stop-EYFP line, and found that tamoxifen treatment resulted in EYFP expression in non-myelinating Schwann cells and myelinating Schwann cells in the sciatic nerve and satellite glial cells in the DRG. The use of PLPCreER to drive reporter expression in satellite glial cells was also reported by Kim et al., 2016 (PMC5017920). Thus the in vivo experiments are not Schwann cell specific. The authors need to restate their results and conclusions taking this into account.
2. The authors use the PLP promoter for their cellular/molecular experiments, but then switch to the P0 promoter for their behavioral experiments. Why do they switch for the behavioral tests? It is not explained. Do the authors have data for behavioral experiments performed using the PLP promoter?
3. Behavioral test should be done in control and HDAC8KO without injury to determine if deletion of HDAC8 in satellite glial cells and schwann cells impacts baseline sensory and motor functions. This is important, since satellite glial cells are known to play a role in pain thresholds.
4. The evidence that sensory function recovery is faster than motor recovery in HDAC8KO mice is supported only by the toe pinch test. Additional behavioral test tailored to sensory functions should be performed to support this observation. Skin re-innervation could be another independent assay to show if regeneration is faster in the absence of HDAC8 in SC.
5. Additional evidence is needed to support the observation that HDAC8 is a specific marker of SC associated with sensory axons. Relying on SCG10 and gap43 is not sufficient. Shin et al 2014 (PMC3947015) reported that SCG10 immunoreactivity rarely colocalizes with the YFP-positive regenerating motor axons, but it is not totally exclusive. Furthermore, whether this is also the case in uninjured nerves has not been examined. The authors should perform staining for HDAC8 and SCG10 in sensory nerves (sural nerve) and in the ventral root that contains only motor axons to ensure SCG10 specificity and provide additional support that HDAC8 is specific for sensory SC.
6. It is not clear if the images in figure 1f is from naïve or injured nerves. As reported in Shin et al 2012, the levels of SGC10 in uninjured nerves are very low and accumulates at the tip of injured axons.
7. There are numerous experiments that lack the proper controls to make accurate conclusions. It seems essential to be able to compare all results with control or sham animals. For example, the authors state for figure 1 that “the regeneration process after lesion was not impaired by the ablation of HDAC8, but instead was accelerated”. However, this conclusion is only based on 1 timepoint at 12 dpl. The authors need to show an earlier timepoint to conclude that regeneration was in fact accelerated. Another example, in figure 3g, the authors manipulate two variables (mouse genotype and injury), but only quantify HIF1a after injury. Without the baseline control, they cannot conclude whether differences in H1F1a result from HDAC8 KO, from the injury, or from both. In 3f, vehicle control should be included along the MG132 treatment. In 3h, nuclear fraction should be shown with cytoplasmic fractions.
8. In Figure 1f and 4g, the authors state that HDAC8 is only expressed in Schwann cells surrounding sensory neurons, or that TRAF7 is expressed in all Schwann cells, based off of one IHC section. To make such conclusions, the authors need quantification and a larger n.
9. The mass spectrometry experiment to identify HDAC8 partners needs more details. How many peptides were identified and why was Traf7 selected for further validation. In all blots presented in Figure 4, molecular weight should be indicated since 4 isoforms are presented. Especially when size of blot is different, such as 4f vs 4h. In panel 4d, the co-IP should be also blotted for Hdac8 and WB for traf7 and hdac8 should be done for the input. The legend needs to state what percentage of the input is loaded to reflect the co-IP efficiency.
10. The conclusions are for the most part too assertive, while the results are based on a single type of experiment and do not hint at other possibilities or openings.

Minor Comments:

1. Why didn't the authors use the HDAC8 KO mice for SC cultures instead of the rat SC culture followed by shRNA?
2. Figure 3: Needs to be reorganized in a way the reader can follow. Experimental design schematics are needed. Conditions are not stated (e.g., is 3g at 1 dpl)?
3. The manuscript flow could be improved to facilitate understanding. The extensive number of supplements make it difficult to navigate between the figures and results.

Reviewer #3

(Remarks to the Author)

The paper by Hertzog et al examines the role of HDAC8 in controlling the regenerative capacity of Schwann cells. The work identifies a link between HDAC8 and the transcription factor cJun, through TRAF7 and HIF1-alpha.

The paper is nicely written and the data is generally well done, but there are many major issues that still need to be addressed.

Major Items:

1. Fig. 1: The authors should explain why the shift from using the PLP-CR-ER system to the P0-CRE-ER. Why look at myelin breakdown etc in one model and regeneration in another? I think to do with the levels of loss of HDAC8 following tamoxifen injection, which are not that impressive (perhaps only approx. 40% down-regulation, not 'efficient' as stated on line 116). The reader presumes that developmental ablation of HDAC8 leads to a defect in PNS development - If this is the case, it should be stated for clarity. Presumably, no destabilisation of the myelin occurred with HDAC8 loss?
2. Fig 1: While there seems to be a sensory deficit, one sensory test is not enough to say this for certain. Other tests (Hargreaves, Von Frey etc), must be used for confirmation of sensory changes or improvements in regeneration. While several timepoints are tested, the data spread on the control animals is huge for the toe pinch tests.

3. Fig. 1: This reviewer is surprised that there is still so much myelin remaining in the control nerve in panel C. At 12 days post-crush injury, can it really be said that these are all sensory fibres that are regenerating more quickly (ie less than 2 microns); surely the motor axons are regrowing still at this early point and maturing in their diameter? Also, given that this is a single timepoint, it's not convincing that these are 'Remak axons' here. If more sensory testing had been done, this would be a more confident conclusion. The reader can also see that this is a transient effect for the toe pinch test; it's gone at 30 days.
4. Fig. 1: If the authors are saying that the myelin debris is being cleared faster, then processes of myelinophagy and immune cell numbers must be carefully examined. Macrophages were stained in some sections (Fig. 2), images shown, but never quantified. Why was this?
5. Fig 1: For the HDAC8/Stathmin stain at 1 day post-injury, the staining is not of very high quality. The Stathmin looks to be as much in the Schwann cells as in the axons and the HDAC8 in both axons and Schwann cells; confusing.
6. Fig. 2: The data with cJun and P-cJun is nice, but I don't see the need to switch between using 1% O₂ hypoxia and a chemically-induced hypoxia (cobalt chloride). The duration of hypoxia seems to change between experiments too (5 or 16 hours; why?). As stated above, why are macrophages stained in panel b? I'd really like to see the confirmation of HDAC8 KD in the main figure. Also, confirmation of HIF1a up-regulation using both paradigms in this figure would be really useful for the reader. Else it seems like too much is assumed here without trying to dig around to find the information.
7. Fig. 3: There are lots of pairwise western blots, which would really be better shown as blots run together. For example, couldn't panels 3b and 3c be combined to show the relative effects of CoCl₂ and 1% hypoxia? There appears to be no confirmation of HIF1a KD anywhere in the paper. Generally I'd like to have seen 2 shRNA constructs trialed in this kind of data to confirm specificity.
8. Fig. 3: Panel i- Is this P-JNK1/2 that the authors are blotting for here?
9. Fig. 3: Panel b – for the graph, two of the data points are actually off the scale off the y-axis.
10. Fig. 4: Panel e – it does seem that it is the smaller isoform that is preferentially targeted by the shRNA – can the isoforms be separately quantified?
11. Fig. 4: Panel g – The HDAC8 stain doesn't look like it is only in 'sensory Schwann cells' If you expect to see HDAC8 in Remak bundles here (I'm presuming this is intact nerve), then the staining doesn't look as if this is true. Perhaps some myelin/GFAP counterstaining could be of use here?

Version 1:

Reviewer comments:

Reviewer #1

(Remarks to the Author)

I congratulate the authors on the thoroughness of their additions to the manuscript in light of my comments. I am satisfied that they have addressed my concerns to the fullest extent that is practically possible. I am happy to recommend this article for publication.

Two additional comments (these are just for academic debate and should not be added to the manuscript)

- 1) Apologies with the comment re Babetto 2020 - I agree they didn't show less S6Kinase in their Hif1 alpha flox mouse. My mistake.
- 2) Regarding Jun phosphorylation. Parkinson et al., 2008 showed that Schwann cells from JunAA mice (without the 2 major active phosphorylation sites, Ser 63 and 73) demyelinate like wild type Schwann cells (Fig 6, particularly 6I and J). Obviously this doesn't rule out a role for phosphorylation at other sites Thr 91 and Thr 93. I would disagree with the authors interpretation of figures 3 and 4 in Parkinson et al 2008. In figure 3 Jun Ala (which has all 4 phosphorylation sites inactivated) inhibits Krox20 induction of PZero showing that Jun N terminal phosphorylation is not required for it to act as a negative regulator of myelin genes in cultured Schwann cells. In figure 4 MKK7 expression inhibits Krox20 induction of myelin genes most likely by re-expression of Jun protein (though I accept a role for Jun phosphorylation is not ruled out in this experiment).

Reviewer #2

(Remarks to the Author)

Hertzog et al., have addressed many of the major concerns and improved the manuscript. However, there are still some minor experimental and editorial adjustments that could further strengthen the study and provide more support to their conclusions.

Comments:

Fig. 1. Thank you for adding before lesion controls to the behavioral tests and adding another measure for mechanical sensitivity (Von Frey test). As presented, the experimental design and statistical analysis performed is confusing. Were the toe pinch, Von Frey, and rotarod tests performed as between-groups or within-groups experiments? Since the statistics used t- tests for each individual time point, each time point likely is from a different cohort of animals. If that is the case, the line graphs should be converted back to bar graphs. However, if each time point was performed on the same cohort of animals, a repeated measures ANOVA should be performed instead of individual t-tests.

Supplementary Figure 4. Thanks for adding the intraepidermal nerve fiber experiment. Can the author explain why they used Neurofilament, rather than PGP9.5, an established marker for skin reinnervation assay?

Fig. 1&2, Supp 11 and 12. One of the authors main conclusions is that HDAC8 is specifically expressed in sensory SCs and controls SCs repair properties and axon regeneration. However, there are no experiments that directly examine co-

localization of HDAC8 with a SCs marker in the nerve. Only close association of HDAC8 positive cells to Stmn2 positive or neurofilament positive axons is shown. While the in vitro experiments and the conditional KO with cre lines are supporting a role of HDAC8 in SC, an in vivo validation that HDAC8 co-localizes with SCs markers around sensory axons and not motor axons should be provided. Such characterization of HDAC8-positive cells could help the reader understand the proportion of SCs expressing HDAC8 in the various nerves.

Additionally, images shown in supplementary figures 11 and 12 could be more convincing. Resolution of Figure S11 as presented is not sufficient; there appears to be strong co-localization of HDAC8 with stathmin 2, and the HDAC8 positive cells surrounding these axons is not clear. In Figure S12, ventral root and sural nerve should be shown at the same magnification. Although HDAC8 intensity is highest in the sural nerve, it appears that a fraction of HDAC8 positive cells in the ventral root does not colocalize with axons and stathmin 2 staining does not appear entirely axonal in the sural nerve. Fig 1f and Supplementary Fig. 10. The authors claim that consistent with faster sensory function recovery, there is an increase in the length of sensory specific STMN2-labeled axons length, but not in the general GAP43-labeled axons. The images provided seem to reflect higher density of GAP43 positive axons past the lesion site in the KO compared to WT. It would be valuable to quantify axon density in addition to axon length, to evaluate the number of axons regrowing in addition to how far they grow, for both Stmn2 and Gap43.

Reviewer #3

(Remarks to the Author)

The authors have responded satisfactorily to my review points.

REVIEWER COMMENTS

Reviewer #1 (Remarks to the Author):

This body of work has been performed by the Jacob lab who are leaders in the roles of HDACs in the nervous systems and the molecular regulation of the Schwann cell phenotype during nerve injury. In their current work they demonstrate that HDAC 8 in Schwann cells partly inhibits JUN upregulation, a central regulator of nerve injury, and this is achieved through TRAF7 and HIF1alpha. What is very intriguing about the study is that Schwann cell HDAC8 appears to be a break for sensory axon regeneration. The authors postulate that this may identify differential regulation of the Schwann cell repair phenotype in sensory versus motor associated Schwann cells. While the data presented in this manuscript are generally solid the experiments don't go quite far enough to support some of the authors claims in the abstract and the discussion. I do believe with the below additions this paper would be a substantial contribution to the literature.

Answer: We want to thank very much Reviewer #1 for the very positive general comments and for suggesting amendments to improve our work. We have explained below after each comment how we implemented the suggested changes and the results of the additional experiments we have carried out.

I suggest the following amendments:

1) In the abstract and line 35/36 and Line 324 – A central claim of this paper is that the conversion into repair cells is controlled by different mechanisms in sensory and motor Schwann cells and the findings in the HDAC8 cKO identify this. The authors never actually look at whether motor Schwann cells phenotype is affected in the HDAC8 cKO. I would suggest the authors need to substantiate this claim by quantifying % Jun +/p-Jun +ve Schwann cells in sensory vs motor Schwann cells at 1 day post injury in HDAC8 KO nerves compared to Wt. Otherwise, the authors need to substantially soften this conclusion in the abstract and the discussion.

Answer: This is a very good point, however almost impossible to show to our point of view. Indeed, Stathmin-2 expression in sensory axons is rapidly downregulated distal to the lesion site (Shin et al., PNAS, 2012; Shin et al., Exp. Neurol., 2014), and this experiment needs to be done at 1 dpl, when c-Jun is already upregulated and many Schwann cells are still in contact with axons and have not yet dramatically changed their morphology, and this needs to be done by cross section. There are some remnants of Stathmin-2 expression at 1 dpl, so what we can do is to quantify c-Jun levels in SCs in contact with axons that still express Stathmin-2 at 1 dpl, but we cannot be sure that axons which do not express Stathmin-2 at 1 dpl are motor axons, they can simply be sensory axons that have lost Stathmin-2 expression. We have tried to stain motor axons with choline acetyltransferase (ChAT), but this did not work. We think that it did not work because ChAT accumulates at the neuromuscular junction, so at the terminal end of motor axons. And we did not find any other useful marker of motor axons or a better marker of sensory axons than Stathmin-2. In any case, we show now in Figure S14 of our revised manuscript that there is a strong increase of c-Jun levels in Stathmin-2-positive axons in HDAC8 KO sciatic nerves compared to control nerves, which strengthen our conclusions. In any case, to be totally correct, we have changed in the abstract and discussion the term “demonstrate” and “show” by “strongly suggest”, which to our point of view is an

understatement given the amount of data provided that support this conclusion, but probably an acceptable compromise.

2) Fig1F and Fig S4. Another central findings is the HDAC8 positive Schwann cells only surround sensory axons, however the authors never quantify this. Can the authors quantify the number of HDAC8 +ve SCs surrounding stathmin 2 +ve and -ve axons.

Answer: We have added the quantification of the percentage of HDAC8-positive SCs surrounding Stathmin2-positive and -negative axons in Fig. S11: 96 ±3 % surround Stathmin2-positive axons and a very small % surrounds Stathmin2-negative axons. It is possible that not all sensory axons express Stathmin2, as shown in our Stathmin-2 staining in sural nerves (Figure S12), which exclusively consists in sensory fibers. Therefore, it is uncertain whether this small number of Stathmin2-negative axons surrounded by HDAC8-positive SCs are sensory or motor axons.

3 a) Fig.S1a – shows 1/3rd reduction in HDAC8 levels by WB in PlpCreERT2 HDAC8 KO and Fig.S1b – shows a 50% reduction in HDAC8 levels by WB I MPZCreERT2 HDAC8 KO. Of course, some of the remaining HDAC8 may be in axons. Presumably axonal HDAC8 is unaffected in the mutants?

Answer: Multiple and extensive tracing studies showed that both PlpCreERT2 and MpzCreERT2 lines that we have used are tight and highly specific, i.e. the PlpCreERT2 line leads to recombination in SCs and oligodendrocytes but not in neurons or other cells of the sciatic nerve (See Leone et al., Mol Cell Neurosci, 2003; Genoud et al., Am J Pathol, 2008; Goebbels et al., J Neurosci, 2010; Delaunay et al., J Neurosci, 2008: no recombination in neurons starting from E13.5; Dumas et al., Glia, 2015), and the MpzCreERT2 line leads to recombination in SCs only and not in neurons or other cells of the PNS (See Leone et al., Mol Cell Neurosci, 2003; Ribeiro et al., Cell Rep, 2013; Stierli et al., Development, 2018; Gomez-Sanchez et al., J Neurosci, 2017). We can therefore be very confident that recombination does not occur in neurons. Another important point to consider is that recombination efficiency using these two CreERT2 lines does not reach 100%: We and others have reported recombination efficiency in a broad range of 15 to 90% (Brügger et al., PLoS Biol, 2015; Ribeiro et al., Cell Rep, 2013; Gerber et al., elife, 2020; Leone et al., Mol Cell Neurosci, 2003). In any case, we have dedicated Figure S2 of our revised manuscript to this point, where we looked at the recombination efficiency of HDAC8 in both PlpCreERT2 and MpzCreERT2 lines. We can see that a fraction of axons express HDAC8 in both control and HDAC8 KO nerves. We can also see in Figure S12 of our revised manuscript that in ventral roots, which are full constituted of motor fibers, SCs do not express HDAC8 but many axons do.

3 b) Related to 3 a) The PlpCreERT2 tends to flox out more in Remak Schwann cells than myelin ones whereas the MPZCreERT2 wont flow out in any Remak/unmyelinated Schwann cells. I think the authors need to make it clear to the reader what % of recombination there is in myelinated and unmyelinated and motor versus sensory (ideally sensory myelinated and sensory unmyelinated by I appreciate this may be difficult to do).

Answer: Reviewer #1 is right, the PlpCreERT2 line leads to recombination in all SCs including myelinating and Remak SCs, whereas the MpzCreERT2 line recombines only myelinating SCs (Ribeiro et al., Cell Rep, 2013). To answer the question of % of

recombination in myelinating versus unmyelinating SCs and motor versus sensory SCs, we carried out the following immunofluorescence in uninjured adult sciatic nerves of PlpCreERT2-HDAC8 KO and control and MpzCreERT2-HDAC8 KO and control, which are presented in Figure S2 of our revised manuscript:

- HDAC8/Stathmin-2/P0
- HDAC8/Isolectin B4 (marker of C-fibers= Remak axons)/Neurofilament

We found in control nerves that around 90% of SCs surrounding IB4+ axons express HDAC8, and around 60% of SCs surrounding myelinated Stathmin-2-positive axons express HDAC8, and 0% of SCs surrounding motor axons (see ventral root staining in Figure S12) express HDAC8. In the PlpCreERT2-HDAC8 KO nerves, we found that 4 to 15% of SCs surround IB4+ axons express HDAC8 (recombination efficiency between 77 and 95%), and that 0 to 12% of SCs surrounding myelinated Stathmin-2-positive axons express HDAC8 (recombination efficiency between 80 and 100%). In the MpzCreERT2-HDAC8 KO nerves, we found no recombination of SCs surrounding IB4+ axons, and that 3 to 17% of SCs surrounding myelinated Stathmin-2-positive axons express HDAC8 (recombination efficiency between 72 and 95%).

This is consistent with previous studies using these Cre lines.

4) Related to the above point the authors don't discuss the different types of sensory axons and Schwann cells. There are A alpha and A beta large myelinated sensory fibres, A delta thinly myelinated sensory fibres and unmyelinated C fibres. Is HDAC8 involved in all or some of these? If the authors are unable to provide data to investigate this then I would accept discussion of this point

Answer: This is indeed quite difficult to do. We can specifically label a fraction of C-fibers with Isolectin B4 (IB4), but to our knowledge there is not a specific marker for each type A-alpha, A-beta and A-delta myelinated sensory fibers. Therefore, we can provide only information of SCs in contact with IB4-positive Remak axons and in contact with sensory myelinating fibers labeled by Stathmin-2.

We show in Figure S2 that around 90% of SCs surrounding IB4+ non-myelinated axons express HDAC8 and that around 60% of SCs surrounding myelinated Stathmin-2 positive axons express HDAC8. This means that a large fraction of sensory fibers (but not all), myelinated and non-myelinated, are surrounded by HDAC8-positive SCs. We do not see a way to distinguish between the different types of myelinated sensory fibers, because of the lack of a specific marker for each subtype. We have added the following text to page 8 of our revised manuscript:

“Of note, HDAC8 was not expressed in all SCs surrounding sensory axons, but was expressed in around 90% of SCs surrounding Isolectin B4 (marker of Remak axons)-positive axons and in around 60% of SCs surrounding myelinated Stathmin-positive axons (**Figure S2**), and HDAC8 was also expressed in subsets of axons (**Figures 1g and S12**). Whether SCs expressing HDAC8 specifically surround A-alpha, A-beta, and/or A-delta myelinated axons remains to be determined, but this analysis will require to identify specific markers for each of these sensory myelinated axon subtypes.”

5) Where is the uninjured analysis of Control and HDAC8 cKO for both PlpCreERT2 and MPZCreERT2— axon counts, g-ratio, SC numbers? Please add this to supplemental.

Answer: We have added images and quantification of degenerated myelin rings, axons (myelinated and non-myelinated) and g-ratio in uninjured contralateral sciatic nerves of Control and HDAC8 KO mice, for PlpCreERT2 at 5, 12 and 30 days post lesion, in Fig. S6 of our revised manuscript. The MpzCreERT2 line was only used for functional analyses, so

ultrastructural data were not presented in our initial submission. The reason for using the PlpCreERT2 line for molecular and ultrastructural analyses and the MpzCreERT2 line for functional recovery is the following: In the PlpCreERT2 line, using a protocol of 2 mg tamoxifen per day for 5 consecutive days, myelinated and non-myelinated SCs are recombined, whereas in the MpzCreERT2 line, myelinating SCs are recombined but recombination in Remak SCs is not efficient. This is why we prefer to use the PlpCreERT2 line for most analyses. However, the PlpCreERT2 line is not suitable for functional recovery analyses because recombination also occurs in oligodendrocytes and it is therefore not possible to know whether the observed phenotype is due to recombination in SCs or in oligodendrocytes or in both. As a side note, HDAC8 is expressed in oligodendrocytes of adult mice. The MpzCreERT2 line leads to recombination only in SCs and not in oligodendrocytes, therefore this line is more suitable for functional recovery analyses. We did collect Control and HDAC8 KO sciatic nerves of the MpzCreERT2 line at 6 months post tamoxifen injection for EM analyses. We have added these data in Fig. S7 of our revised manuscript.

6) Fig. 1C Would be better to quantify demyelination at one or two earlier time points (say day 3 and 7) as 12 days is very late.

Answer: We have added the quantification of intact myelin rings at 5dpl in Figure 1C. At this time point, we found that there are less intact myelin rings in the HDAC8 KO nerves compared to control nerves, indicating more efficient demyelination in the absence of HDAC8.

7) Fig. 1 Would be better to display functional recovery data in line graphs with data points connected by lines over time. It is odd that the time points start at day 12 post injury. Better to plot pre-lesion tests (was there a difference between control and KO uninjured?) and then there should be a few earlier time points prior to visualising regeneration in WTs or mutants – Firstly to show that nerve crushes were complete between genotypes and biological replicates and secondly to demonstrate that there is precocious axon regeneration.

Answer: In case no toe exhibited sensitivity, the same test was applied to toes of the contralateral side (uninjured side), which always resulted in immediate withdrawal. We have written this sentence into the material and method section of our manuscript. We have also added for all behavioral tests the data before lesion, at 3 days post lesion (dpl), and at 5 dpl. The additional time points for the toe pinch test are now included in our revised manuscript in Figure 1. Before lesion and at 3 dpl, there is no difference between HDAC8 KO and controls, but starting at 5 dpl, there is already a small but significant increase of sensory function recovery in the HDAC8 KO mice compared to Control mice. We have verified this timing of sensory function recovery with a von Frey test. For the Rotarod test and Inverted grid test, we have also added the time points before lesion and at 5 days post lesion. We have also connected the time points by lines when possible. Of note, the saphenous nerve is not injured during the protocol of sciatic nerve crush lesion, and this nerve innervates a part of the dorsal hind foot (see for example Hsieh et al., J. Pathol. Exp. Neurol., 2000), so it is not possible to have a zero score reaction at 3 dpl after sciatic nerve crush lesion. We can see however that all animals, controls and HDAC8 KO display a similar residual sensory function at 3 dpl.

8) Fig. 1d. The display of the data for the toe pinch is odd as all the toes are grouped together in the display yet the time to regeneration of each toe is slightly different – digit 1 being the fastest (day 13) and 5 the slowest (day 16) – see Siconolfi and Seeds 2001, DOI: <https://doi.org/10.1523/JNEUROSCI.21-12-04348.2001>. It would be better to display the

data for each individual toe as I've outline in point 3. Some of this data could be put in supplemental.

Answer: We have also done this analysis ourselves and observed a similar pattern of recovery, such as in Siconolfi and Seeds, 2001. We have added this information to Figure S3 of our revised manuscript.

9) Fig 1d. I would suggest an additional method to test functional sensory recovery between control and HDAC8 and cKO such as Von Frey hair and/or Hargreaves test if available.

Answer: In addition to the toe pinch test, we have carried out a von Frey test, which confirmed faster sensory recovery of the HDAC8 KO mice compared to the Controls. We have added these data to Figure 1e.

10) Fig S7 the down regulation of c-Jun in Hif1alpha flox nerves at 3 days is fairly slight. There are only 3 data points for the mutant – 2 of which are equivalent to the control and one animal with 50% the JUN level. Given that this reported finding is a key conclusion that the proposed mechanism hinges on I would suggest the authors increase the number of mutants to 5 and do additional time points after injury say 1 or 2 days as well as 5 or 7 days post injury. Also, this data should be in the main figure panels not in supplemental.

Answer: Here, we show that the ablation of HDAC8 in Schwann cells leads to increased levels of HIF1a, leading to increased c-Jun and phospho-c-Jun levels. HIF1a is widely known to induce c-Jun upregulation and phosphorylation. We did not aim to prove this point because others have done that before. Our aim was not either to analyze the functions of HIF1a after injury, but to show the functions of HDAC8 after injury. Our findings show that ablation of HDAC8 leads to increased HIF1a, which, as already known, leads to increased levels of c-Jun and phospho-c-Jun. We did not intend to prove that HIF1a is necessary to induce c-Jun expression after injury, but we intended to show that when HIF1a levels are increased (here by HDAC8 ablation), c-Jun is upregulated and more phosphorylated. The upregulation and phosphorylation of c-Jun in Schwann cells after lesion can be independent of HIF1a in wild type mice, but if we increase the levels of HIF1a (here by HDAC8 ablation), we also increase c-Jun and phospho-c-Jun levels. Here, we just wanted to explain why we thought about applying hypoxia to our cell cultures. To our point of view, this piece of data belongs to the supporting additional data and not to the main findings. We had initially received two additional HIF1a KO, but one of them did not show any recombination, while the other one showed modest recombination efficiency.

We have now paired the animals by sex and by gel, and quantified 4 HIF1a KO samples compared to their respective control samples. We can see that all KO express lower levels of HIF1a and c-Jun compared to their respective controls (Figure S17 of our revised manuscript).

We unfortunately do not have access to more HIF1a KO animals, so we could not run additional time points. As explained above, we do not feel that this is a major piece of data in our study and would like to ask Reviewer #1 for his/her understanding in this situation where we cannot get more animals. We are ready to totally remove these data from the manuscript if Reviewer #1 finds this a better solution.

11) Related to Fig S7 What is the phenotype of the Hif1alpha cKO mouse – does it have slower myelin clearance/regeneration? As Hif1alpha is proposed as a key regulator of JUN it is important to show this data.

Answer: As explained above, we do not propose that HIF1alpha is a key regulator of c-Jun, this is already known. And we also do not propose that HIF1a is a key inducer of c-Jun in SCs after injury. It is possible that c-Jun is upregulated by different mechanisms than HIF1alpha in SCs after injury in wild type mice. What our data show is that ablating HDAC8 leads to increased levels of c-Jun and phospho-c-Jun through increased HIF1a levels. We do not think that the phenotype of the HIF1a KO mice belongs to the study of the phenotype of HDAC8 KO mice. Describing the phenotype of HIF1a KO mice would be a tremendous amount of additional work and we do not think that this is necessary to support our conclusions. To our opinion, this would be a different study that needs to be properly and completely conducted independently to the study of HDAC8 functions in SCs. In addition, as mentioned above, we do not have access to more HIF1a KO animals.

12) Discussion + Fig5. A lot of weight is put on the role of JNK and JUN phosphorylation in the authors proposed mechanism. The authors need to discuss the evidence for JNK and Jun phosphorylation in nerve injury. My understanding is N terminal phosphorylation is not required based on Parkinson 2008 doi/ JCB 62510.1083/jcb.200803013 and data supporting a role for JNK in nerve injury is solely from in vitro studies.

Answer: Here again, similar to comments 10 and 11, we do not claim that JNK and c-Jun phosphorylation are needed for the regeneration process after sciatic nerve crush injury. We however claim that if the levels of phosphorylated JNK, phosphorylated c-Jun and total c-Jun are increased, here by the ablation of HDAC8, there is faster conversion into the SC repair phenotype. The lab of Kristjan Jessen has shown that both c-Jun and phospho-c-Jun are equally capable of inducing SC demyelination: See Figure 3A,3B,3G of Parkinson 2008 doi/ JCB 62510.1083/jcb.200803013. In addition, in Figure 4 of the same article, Parkinson et al. show that the overexpression of MKK7, which specifically phosphorylates JNK, leading to increased levels of c-Jun phosphorylation and also of c-Jun, induces c-Jun-dependent SC demyelination. To make that clearer, we have added discussion on this point.

Additional points:

A) Introduction: In addition to Shh add in mention of additional 2 known regulatory mechanisms of c-JUN in Schwann cells after injury - mTOR regulates c-JUN protein levels – Norrmén et al., 2018 DOI:10.1523/JNEUROSCI.3619-17.2018 and Kim et al., 2018 O-GlcNAcylation of JUN and inhibits transcription activity allowing remyelination doi/10.1073/pnas.1805538115

Answer: We have added these two references in the Introduction. Many thanks for pointing that out.

B) Line 119 – These are not Remak Schwann cells/axons but regeneration clusters

Answer: Many thanks, we have changed the text accordingly.

C) Figure 1 C – For EM analysis – how many sections per nerve were quantified and which fascicle of the sciatic was quantified (peroneal/tibial/sural) or where all 3 quantified.

Answer: All EM analyses were carried out in the full sciatic nerve, at 5 mm distal to the lesion site. We have added this information in the manuscript in the material and methods

section. Three sections per animals were analyzed for the number of degenerating myelin rings in semithin sections, and 5 images taken in three sections per animal were analyzed for ultrathin sections (all axons in these 5 images were counted). This information has been added to the manuscript, in the figure legends.

D) Fig 1F – Is this a maximum projection or individual z position?

Answer: This is a maximum projection of z-stacks. We have added this information in the Figure legend of our revised manuscript. Thank you for pointing that out.

E) Fig 3i the WB for p-JNK does not seem representative of the quantifiable difference (a 25% reduction)

Answer: All n look the same. The apparent difference between the image and the quantification is due to a relatively high background. We have removed the background and re-quantified. The new quantification shows a decrease of 85% of p-JNK, which seems very close to what we can see in the image. Thank you for pointing that out.

F) Discussion: Authors should discuss the findings of Babetto et al., 2020. They floxed HIF1alpha out of Schwann cells in nerve injury and showed less phosphorylated S6 kinase and slightly increased axon disintegration after injury at 36hours.

Answer: We have looked carefully at the findings in Babetto et al., 2020, and we could not find data showing less phosphorylated S6 kinase in HIF1a KO mice. Concerning the slightly increased axon disintegration at 36 h post injury, they have seen that only when HIF1a is ablated with a P0Cre line, so developmentally. To ablate HIF1a in adult SCs, they used the iSox10CreERT2 line. With this line, they did not see any increased axonal disintegration. See Figure 7g. We have anyway references the article of Babetto et al., 2020, in our manuscript.

G) In general authors should try to use colourblind friendly colour combinations for immunos – avoid red/green

Answer: We have used green, magenta and blue when we need to show only 3 channels, but when we need to show 4 channels, we do not know which other combination than green, magenta, blue and red to use. This concerns Fig. 1g, Fig. 4g, Fig. S2, Fig. S11, Fig. S12, Fig. S14). However, to make it possible for colourblinding people to understand the results, we also show the green and red channels separately in addition to the merged channels. We hope it is acceptable like this.

Reviewer #2 (Remarks to the Author):

Hertzog et al. investigated the role of HDAC8 in the conversion of sensory Schwann cells into repair cells under hypoxic conditions. They first show that HDAC8 expression is upregulated in peripheral nerves after lesion. They then show that ablation of HDAC8 in adult Schwann cells results in less myelin debris and more axons per surface area at 12 dpl. Using primary rat SCs cultured under conditions that promote conversion of SCs into the repair phenotype, they show that downregulation of HDAC8 led to increased levels of c-Jun and phosphorylated c-Jun. These effects were more pronounced when cells were incubated under hypoxic conditions. Therefore, the remaining studies were performed in hypoxic conditions. The authors showed that HDAC8 downregulation increased HIF1a levels in both

normoxic and hypoxic conditions. They hypothesized that HDAC8 regulates HIF1a stability via a VHL-independent mechanism. They also suggest that downregulation of HDAC8 in SCs leads to HIF1a- and JNK-dependent c-Jun phosphorylation and upregulation. Lastly, they investigated the potential binding partners of HDAC8 by mass spectrometry. They identified TRAF7 as a putative binding partner HDAC8. They conclude that HDAC8 regulates the short and long TRAF7 isoforms, leading to ubiquitination and degradation of HIF1a and inhibition of the JNK pathway. The results suggest an interesting mechanism for conversion of Schwann cells into repair cells under hypoxic conditions. However, there are some major concerns regarding the lack of controls in the experiments, and conclusions stated by the authors that are not sufficiently substantiated by the data presented. Furthermore, the figures are difficult to understand without schematics for each experimental design, and the manuscript would benefit from editing to improve flow and clarity. This manuscript needs significant experimental and editorial revisions to be suitable for publication in Nature Communications.

Answer: We want to thank Reviewer #2 for finding our study interesting and for suggesting changes to improve our manuscript.

Major Comments:

1. The authors state that their HDAC8 KO mouse line is specific to Schwann cells. However, PLP and P0 are expressed in satellite glial cells. Elbaz et al 2022 (PMC9354470) used the PLP-CreERT-ROSA26-Stop-EYFP line, and found that tamoxifen treatment resulted in EYFP expression in non-myelinating Schwann cells and myelinating Schwann cells in the sciatic nerve and satellite glial cells in the DRG. The use of PLPCreER to drive reporter expression in satellite glial cells was also reported by Kim et al., 2016 (PMC5017920). Thus the in vivo experiments are not Schwann cell specific. The authors need to restate their results and conclusions taking this into account.
2. The authors use the PLP promoter for their cellular/molecular experiments, but then switch to the P0 promoter for their behavioral experiments. Why do they switch for the behavioral tests? It is not explained. Do the authors have data for behavioral experiments performed using the PLP promoter?
3. Behavioral test should be done in control and HDAC8KO without injury to determine if deletion of HDAC8 in satellite glial cells and schwann cells impacts baseline sensory and motor functions. This is important, since satellite glial cells are known to play a role in pain thresholds.

Answer: We thank Reviewer #2 for this comment. We have now run all our behavioral tests before lesion (data are added in Figure 1d of our revised manuscript) and we can see that there is no difference between control and HDAC8 KO mice before lesion, suggesting that potential recombination in satellite glial cells does not impact our results. For all behavioral analyses, we have used the P0CreERT2 line and not the PlpCreERT2 line to avoid looking at potential effects coming from recombination in other glia such as oligodendrocytes. For molecular and ultrastructural analyses, we however use the PlpCreERT2 line because recombination occurs in all SCs, whereas the P0creERT2 line leads to recombination in myelinating SCs only. We have now explained that in the manuscript. We do not know whether the P0CreERT2 line leads to recombination in satellite glia, we could not find this information, but since the baseline levels of the behavioral tests before lesion are similar in HDAC8 KO and control littermate mice, we think that the behavioral data after sciatic nerve crush lesion can be confidently attributed to recombination in SCs. We have added in the manuscript that the PlpCreERT2 line also leads to recombination in satellite glial cells and

have added the two references mentioned by the reviewers (Elbaz et al., 2022, and kim et al., 2016).

4. The evidence that sensory function recovery is faster than motor recovery in HDAC8KO mice is supported only by the toe pinch test. Additional behavioral test tailored to sensory functions should be performed to support this observation. Skin re-innervation could be another independent assay to show if regeneration is faster in the absence of HDAC8 in SC.

Answer: We thank Reviewer #2 for this suggestion. We have added a von Frey test, which confirms our data with the toe pinch test (data added to Figure 1d) and we have also looked at hind paw re-innervation, which also supports our findings (data added in Figure S4 of our revised manuscript).

5. Additional evidence is needed to support the observation that HDAC8 is a specific marker of SC associated with sensory axons. Relying on SCG10 and gap43 is not sufficient. Shin et al 2014 (PMC3947015) reported that SCG10 immunoreactivity rarely colocalizes with the YFP-positive regenerating motor axons, but it is not totally exclusive. Furthermore, whether this is also the case in uninjured nerves has not been examined. The authors should perform staining for HDAC8 and SCG10 in sensory nerves (sural nerve) and in the ventral root that contains only motor axons to ensure SCG10 specificity and provide additional support that HDAC8 is specific for sensory SC.

Answer: This is a very good point. We have collected sural nerves and ventral roots and stained cross sections for HDAC8/Stathmin-2 8SCG10/Neurofilament, and show in Figure S12 of our revised manuscript that SCG10 is not detected in ventral roots but in contrast is abundantly expressed in sural nerves. In addition, HDAC8 is not expressed in any SCs of ventral roots, but is abundantly expressed in SCs of sural nerves. As a side note, HDAC8 is expressed in many axons of ventral roots. In addition, such as Shin et al. (2012-2014) have found, we show that SCG10 is expressed in uninjured axons of sciatic nerves, and concentrates in growth cones of regenerating axons after sciatic nerve crush injury (data presented in Figure S9 of our revised manuscript).

6. It is not clear if the images in figure 1f is from naïve or injured nerves. As reported in Shin et al 2012, the levels of SGC10 in uninjured nerves are very low and accumulates at the tip of injured axons.

Answer: These images were taken in the proximal part of the injured sciatic nerve at 1 dpl, where SCG10 levels are high (see whole-nerve IF in Figure 1e). However, we found similar staining pattern in contralateral uninjured sciatic nerves. To test whether SCs are associated with sensory axons, we needed to use either uninjured nerves or after lesion in the proximal part of the sciatic nerve. SCG10 is expressed in uninjured nerves. We and Shin et al., 2012 (PNAS, doi: 10.1073/pnas.1216204109, Fig. 1 and Fig. 2A, Fig. 2C-bottom panel) have shown robust expression of SCG10 in uninjured cultured DRG neurons and sciatic nerves by immunofluorescence and WB. We have added our data, too, for confirmation in Figure S9 of our revised manuscript.

7. There are numerous experiments that lack the proper controls to make accurate conclusions. It seems essential to be able to compare all results with control or sham animals. For example, the authors state for figure 1 that “the regeneration process after lesion was not impaired by the ablation of HDAC8, but instead was accelerated”. However, this conclusion is only based on

1 timepoint at 12 dpl. The authors need to show an earlier timepoint to conclude that regeneration was in fact accelerated.

Answer: We have shown whole-nerve immunofluorescence for Stathmin-2 at 3 dpl showing longer regrown axons from the lesion site and electron microscopy at 12 dpl showing increased number of regrown axons at the same distance to the lesion site. With these two time points, we can be sure that axonal regrowth is faster in the HDAC8 KO compared to controls. At 5 dpl, we can see a decreased number of intact myelin rings in the HDAC8 KO compared to controls, indicating a faster demyelination in the absence of HDAC8, which is consistent with faster axonal regrowth. It is very difficult to count regrown axons at an early time point after lesion by EM because there are too many degenerating myelin rings. We think that the best is to use whole-nerve immunofluorescence for early time points and EM for late time points. To make sure that we don't make an overstatement here, we have removed the term "accelerated" and changed to "promoted". Of note, sensory function regeneration is shown at different time points, and we can see there that regeneration is accelerated.

Concerning additional controls, we have added the data of the contralateral nerves for each time point of ultrastructural analyses. For the WB and IF analyses of the HDAC8 KO and controls, all contralateral nerves are also shown.

Another example, in figure 3g, the authors manipulate two variables (mouse genotype and injury), but only quantify HIF1a after injury. Without the baseline control, they cannot conclude whether differences in HIF1a result from HDAC8 KO, from the injury, or from both.

Answer: All controls including contralateral cytoplasmic and nuclear levels were shown in Figure S10. We have now moved these data to Figure 3h of our revised manuscript, so that it is more obvious.

In 3f, vehicle control should be included along the MG132 treatment. In 3h, nuclear fraction should be shown with cytoplasmic fractions.

Answer: We have always run the vehicle controls next to the MG-132-treated cells, but we found it quite redundant to Figure panel 3b. We have however now added the vehicle control to Figure 3f (Figure 3g of our revised manuscript). We have also added the nuclear fraction to Figure 3h (Figure 3i of our revised manuscript), although there is no HIF1a detectable at this time point in the nuclear fraction.

8. In Figure 1f and 4g, the authors state that HDAC8 is only expressed in Schwann cells surrounding sensory neurons, or that TRAF7 is expressed in all Schwann cells, based off of one IHC section. To make such conclusions, the authors need quantification and a larger n.

Answer: We have already stated in the figure legends of our initial manuscript the number of animals that were used for these data: "Three animals were used and a representative image is shown." (legend of Figure 1f), and "Three WT mice were used for this experiment and representative images are shown." (legend of Figure 4g). In addition, we have now included the quantification of HDAC8+SCs surrounding Stathmin-2-positive axons in our revised manuscript in Figure S11 and Figure S2. Concerning TRAF7 expression, we have not found any Schwann cell that does not express TRAF7, so we wrote that TRAF7 is expressed in all

Schwann cells (100% of Schwann cells in all sections of the three WT mice). We cannot add quantification, this means 100%±0.

9. The mass spectrometry experiment to identify HDAC8 partners needs more details. How many peptides were identified and why was Traf7 selected for further validation.

Answer: In Table S1, the number of peptides are indicated: 3 peptides were identified for HDAC8 and 2 peptides were identified for Traf7 (they are highlighted in yellow). The 2 peptides identified for TRAF7 are comprised within the short TRAF7 isoform. We have added this information to the manuscript. We have indicated already in our initial manuscript (page 10, lines 232-236) that among the putative binding partners of HDAC8 with a function in ubiquitin conjugation, we focused on TRAF7 because it is localized in the cytoplasmic compartment in both primary rat SCs (Figure 4b) and adult sciatic nerves (Figure 4c) such as HDAC8, possesses a RING domain with E3 ligase activity and can therefore mediate ubiquitination^{44,45}. We have tested other putative binding partners of HDAC8 but they did not turn out to be interesting for our study. We do not know what else to say, those are really the reasons why we followed on with TRAF7.

In all blots presented in Figure 4, molecular weight should be indicated since 4 isoforms are presented. Especially when size of blot is different, such as 4f vs 4h.

Answer: We have added the size of the bands in the Traf7 WB images in Figure 4.

In panel 4d, the co-IP should be also blotted for Hdac8 and WB for traf7 and hdac8 should be done for the input. The legend needs to state what percentage of the input is loaded to reflect the co-IP efficiency.

Answer: The input always represents 3% of the lysate used for the IP. We added this information to the figure legend. In all our IPs, we have separated the same lysate into two equal fractions, one for the IP of interest and one for the control IP. We have also added this information to the figure legend. Even though the IPs have been carried out on the same lysate separated into two equal fractions, we still have loaded the input and blotted for GAPDH, and also for TRAF7 and HDAC8 in Figure 4d.

10. The conclusions are for the most part too assertive, while the results are based on a single type of experiment and do not hint at other possibilities or openings.

Answer: We hope that Reviewer #2 will find that our conclusions are now strengthened in our revised manuscript with the requested additional experiments, and also the re-wording.

Minor Comments:

1. Why didn't the authors use the HDAC8 KO mice for SC cultures instead of the rat SC culture followed by shRNA?

Answer: In our hands, the culture of primary rat Schwann cells is very robust and the protocols for proliferation, differentiation and for mimicking the conversion into the repair phenotype are very well established in our lab and other labs. The culture of purified primary mouse Schwann cells is more problematic, at least in our hands and in most other labs. For example, mouse Schwann cells cannot be maintained for a long time in purified cultures, the

amount of material is therefore low and the protocols for proliferation and mimicking the conversion into the repair phenotype are not well established.

2. Figure 3: Needs to be reorganized in a way the reader can follow. Experimental design schematics are needed. Conditions are not stated (e.g., is 3g at 1 dpl)?

Answer: All the details were found in the figure legend of our initial manuscript. For example, we wrote that Figure 3g represents crushed sciatic nerves at 1 dpl (day post lesion). To make the conditions and experimental design clearer, we have added them in each figure panel. In addition, all cell culture protocols are described in details in the Material and methods section.

3. The manuscript flow could be improved to facilitate understanding. The extensive number of supplements make it difficult to navigate between the figures and results.

Answer: We have added some of the supplementary data into the main figures to facilitate understanding. We hope the manuscript flow is improved now. However, some additional experiments requested by the reviewers needed to be added to the supporting material. We think it is important to show the main findings in the main figures and the supporting data in the supporting material to not overflow the reader with supporting material. We hope we have now found the right balance.

Reviewer #3 (Remarks to the Author):

The paper by Hertzog et al examines the role of HDAC8 in controlling the regenerative capacity of Schwann cells. The work identifies a link between HDAC8 and the transcription factor cJun, through TRAF7 and HIF1-alpha.

The paper is nicely written and the data is generally well done, but there are many major issues that still need to be addressed.

Answer: We thank Reviewer #3 for the positive comments and for the suggestions to improve our manuscript.

Major Items:

1. Fig. 1: The authors should explain why the shift from using the PLP-CR-ER system to the P0-CRE-ER. Why look at myelin breakdown etc in one model and regeneration in another? I this to do with the levels of loss of HDAC8 following tamoxifen injection, which are not that impressive (perhaps only approx.. 40% down-regulation, not 'efficient' as stated on line 116.

Answer: For all behavioral analyses, we have used the P0CreERT2 line and not the PlpCreERT2 line to avoid looking at potential effects coming from recombination in other glia such as oligodendrocytes. For molecular and ultrastructural analyses, we however use the PlpCreERT2 line because recombination occurs in all SCs, whereas the P0creERT2 line leads to recombination in myelinating SCs only. We have now explained that in details in the manuscript.

The reader presumes that developmental ablation of HDAC8 leads to a defect in PNS development - If this is the case, it should be stated for clarity. Presumably, no destabilisation of the myelin occurred with HDAC8 loss?

Answer: Developmental function of HDAC8 was not the topic of our study that is focused on regeneration. However, we have also generated the *DhhCre-Hdac8* KO and control mice, and have not detected any developmental phenotype: HDAC8 KO animals were indistinguishable from their control littermates. We have also checked developmental expression of the main promyelinating factors and myelin protein and of other class I HDACs (for potential compensation) from P10 to P60, and carried out ultrastructural analyses at P90. No defect was detected. However, this does not exclude a potential transient effect or an effect at a later stage or a compensatory mechanism by another factor. We have anyway added these data to our revised manuscript in Figure S8, but we would have preferred not to since this is not the topic of the current study. Concerning a potential effect of HDAC8 in the maintenance of PNS integrity, there is also no obvious phenotype until at least 6 months post tamoxifen injections: HDAC8 KO and control littermates are indistinguishable, and ultrastructural analyses by EM images do not identify any defect. We have added these data to Figure S7 of our revised manuscript. However, here again, the function of HDAC8 in the maintenance of PNS integrity was not the topic of this study, and these data do not exclude a phenotype at a later stage, so we would have preferred not to add these data to the present regeneration study. In conclusion, no destabilisation of the myelin occurred with HDAC8 loss during either development or maintenance.

2. Fig 1: While there seems to be a sensory deficit, one sensory test is not enough to say this for certain. Other tests (Hargreaves, Von Frey etc), must be used for confirmation of sensory changes or improvements in regeneration. While several timepoints are tested, the data spread on the control animals is huge for the toe pinch tests.

Answer: We have added time points including before lesion, at 3 dpl and 5 dpl and a von Frey sensory test to our revised manuscript. Early time points at 3 dpl show that all animals have been injured to the same levels. There are always differences in behavioral tests in between animals, this is why these tests need to include a high number of animals, here 12 mice per experimental group. We have not excluded any animal from the quantification. One Control animal did not regenerate well, but even if we remove this control animal, the data remain significant, so we think that there is no need to remove any animal from the quantification.

3. Fig. 1: This reviewer is surprised that there is still so much myelin remaining in the control nerve in panel C.

Answer: We do not really know how to answer this comment. This is how a sciatic nerve looks like at 12 dpl, at least in our hands. We have similar images with different studies. We have added a 5 dpl time point showing more remaining intact and degenerated myelin rings than at the 12dpl where no intact myelin ring could be detected (only degenerated ones). In addition, one-to-one relationships between Schwann cells and axons are frequently observed with even a thin myelin sheath, corresponding to the 12 dpl time point. We are very sure that we did not mix up the time points.

At 12 days post-crush injury, can it really be said that these are all sensory fibres that are regenerating more quickly (ie less than 2 microns); surely the motor axons are regrowing still at this early point and maturing in their diameter? Also, given that this is a single timepoint, it's not convincing that these are 'Remak axons' here. If more sensory testing had been done, this would be a more confident conclusion. The reader can also see that this is a transient effect for the toe pinch test; it's gone at 30 days.

Answer: There are sensory fibers of high caliber. Unmyelinated C-fibers range from 0.2 to 1.5 μm in diameter, A-delta myelinated sensory fibers range from 1 to 5 μm in diameter, A-beta sensory fibers range from 6 to 12 μm in diameter and A-alpha range from 13 to 20 μm in diameter. At this time point, we can distinguish regenerating axonal clusters and one-to-one relationships between axons and Schwann cells. Not all axons are remyelinated at this stage and potentially not all axons are in a one-to-one relationship. To make that clear, we have changed the term Remak to regenerating axonal clusters. We have run another sensory test (von frey test presented in Figure 1d) to strengthen our conclusions, as suggested by the Reviewers, which confirm a sensory phenotype. We found an accelerated sensory function recovery, but the control animals also eventually recover their sensory function.

4. Fig. 1: If the authors are saying that the myelin debris is being cleared faster, then processes of myelinophagy and immune cell numbers must be carefully examined. Macrophages were stained in some sections (Fig. 2), images shown, but never quantified. Why was this?

Answer: We have tried 3 different LC3B antibodies to show increased autophagy in the HDAC8 KO nerves compared to controls, but we could not find a clean one. We are showing below the Western Blots with the best antibody we have tried, in contralateral and crushed nerves of HDAC8 KO and control mice at 3 days post lesion, where the levels of c-Jun are increased in the HDAC8 KO samples. We can see in these blots that the amount of LC3B II compared to LC3B I is increased in all HDAC8 KO crushed samples compared to control crushed samples. However, since the signal does not give clear bands, it is difficult to quantify. According to the many articles on c-Jun from the Jessen lab, it is very well accepted that c-Jun is an inducer of myelinophagy in peripheral nerves (e.g. Fazal et al., 2017), so we think it may not be necessary to show the LC3B blots in our study, in particular since we are showing now that there is more efficient myelin breakdown in HDAC8 KO nerves after injury at 5 dpl (decreased number of intact myelin rings in Figure 1c of our revised manuscript). We are still showing below LC3B and GAPDH Western blots on lysates of contralateral and crushed sciatic nerves of 3 different control and 3 different HDAC8 KO mice at 3 dpl, but would prefer to not include these data in our revised manuscript because of low quality of the data and because it is not quantifiable:

We have quantified the number of macrophages at 1 and 5 dpl and included these data in our revised manuscript (Figure S13). There is no difference in macrophage numbers between Control and HDAC8 KO nerves.

5. Fig 1: For the HDAC8/Stathmin stain at 1 day post-injury, the staining is not of very high quality. The Stathmin looks to be as much in the Schwann cells as in the axons and the HDAC8 in both axons and Schwann cells; confusing.

Answer: We show by cross section that HDAC8 is strongly expressed in Schwann cell cytoplasm and at somewhat lower levels in axons. Stathmin-2 is expressed by axons, but not by Schwann cells, this is widely accepted. Indeed, the Schwann cell staining surrounds the Stathmin-2 staining. Stathmin-2 is targeted to the axonal membrane, so the stathmin-2 staining is apposed to the HDAC8 staining of the Schwann cells and can seem to overlap sometimes, however when looking closely, the HDAC8 and Stathmin-2 stainings do not overlap. Figure 1g shows nice examples indicated by white arrows. HDAC8 is also expressed in subsets of axons, we cannot change that. We have added a staining of HDAC8 in ventral roots (fully motor) and sural nerves (fully sensory) in Figure S12. These data show that SCs present in ventral roots do not express HDAC8, whereas many SCs present in sural nerve express HDAC8. In addition, many axons in ventral roots express HDAC8.

6. Fig. 2: The data with cJun and P-cJun is nice, but I don't see the need to switch between using 1% O₂ hypoxia and a chemically-induced hypoxia (cobalt chloride). The duration of hypoxia seems to change between experiments too (5 or 16 hours; why?).

Answer: We wanted to have two different ways to induce hypoxia to make sure that our results are similar in these two different ways. We have mostly used 16 h of hypoxia, but to show early upregulation of HIF1 α in the cytoplasm before reaching the nucleus, we have used 5 h of hypoxia.

As stated above, why are macrophages stained in panel b?

Answer: We have stained for macrophages to eliminate a potential expression in macrophages to make sure that we look at c-Jun and phospho-c-jun expression in Schwann cells.

I'd really like to see the confirmation of HDAC8 KD in the main figure.

Answer: We have added the data showing the efficiency of HDAC8 KD in the main figure (in figure 2c).

Also, confirmation of HIF1 α up-regulation using both paradigms in this figure would be really useful for the reader. Else it seems like too much is assumed here without trying to dig around to find the information.

Answer: In our initial manuscript, we had already presented the HIF1 α upregulation in cells where HDAC8 is downregulated by shRNA for both paradigms, hypoxia by 1% O₂ in Figure 3c, and hypoxia by CoCl₂ in Figure 3b (N and H for normoxia and hypoxia). To make that clearer, we have described the two types of hypoxia directly in the figure, in addition to the figure legend and main text. Or did the reviewer mean just the basic upregulation of HIF1 α in the two paradigms? In case this is what is meant, we have added to Figure 3c the normoxia conditions, so we can see how much HIF1 α is upregulated under 1% O₂ after 16 h. We found that quite redundant to Figure 3b, but of course this shows that all conditions are well controlled.

7. Fig. 3: There are lots of pairwise western blots, which would really be better shown as blots run together. For example, couldn't panels 3b and 3c be combined to show the relative effects of CoCl₂ and 1% hypoxia?

Answer: We don't really understand why these blots should be run together and compared. Those are different conditions, one mimicking the hypoxia state chemically and the other one conducted under low O₂, so these experiments were not done at the same time, so we don't really aim at comparing them. Our aim by using two different hypoxia paradigms was to show that HDAC8 downregulation leads to increased HIF1a, c-Jun and phospho-c-Jun levels in both hypoxia paradigms, so that this effect is robust. Also, we are showing these two experimental paradigms side-by-side or nearby in the same figures, so that it is easy to see the fold increase due to HDAC8 downregulation in both experimental paradigms. We would prefer to keep these data in separated panels for clarity reasons. We have however added to figure 3c the normoxia conditions that were run at the same time and on the same gels as the hypoxia with 1% O₂. This way, we can nicely compare the increase of HIF1a in both paradigms.

There appears to be no confirmation of HIF1a KD anywhere in the paper. Generally I'd like to have seen 2 shRNA constructs trialled in this kind of data to confirm specificity.

Answer: The confirmation of HIF1a KD was presented in Figure S9 of our initial manuscript. We have now moved these data to Figure 3d of our revised manuscript. We have tried to find another HIF1a shRNA that works on rat HIF1a, but we did not find. We think the one we have used is highly efficient and has also been validated by Sigma. In addition, it led to the expected effect of decreased c-Jun-, phospho-c-Jun and phospho-JNK, consistent with the known functions of HIF1a.

8. Fig. 3: Panel i- Is this P-JNK1/2 that the authors are blotting for here?

Answer: Yes, it is. We have added the missing "1/2" to Figure 3i. We thank Reviewer #3 for pointing this out.

9. Fig. 3: Panel b – for the graph, two of the data points are actually off the scale off the y-axis.

Answer: Yes, they are. For esthetical reasons, we initially did not show the full y axis. We have changed that in our revised manuscript in all figures.

10. Fig. 4: Panel e – it does seem that it is the smaller isoform that is preferentially targeted by the shRNA – can the isoforms be separately quantified?

Answer: All TRAF7 isoforms are targeted by the TRAF7 shRNA, with indeed a very high efficiency for the small isoform. We have now separated the quantification of the small and long isoforms in our revised manuscript.

11. Fig. 4: Panel g – The HDAC8 stain doesn't look like it is only in 'sensory Schwann cells' If you expect to see HDAC8 in Remak bundles here (I'm presuming this is intact nerve), then the staining doesn't look as if this is true. Perhaps some myelin/GFAP counterstaining could be of use here?

Answer: Many sensory fibers (A-alpha, A-beta, A-delta) are myelinated, only the C-fibers (Remak) are unmyelinated. For Remak Schwann cells and Schwann cells surrounding myelinated sensory fibers, we have added Figure S2. To stain Remak axons, we have used Isolectin B4 (expressed in Remak axons and endothelial cells), and to stain for sensory

myelinated fibers, we have co-stained for Stathmin-2 and P0. As mentioned earlier, we have also quantified the % of Schwann cells expressing HDAC8 that are associated with Stathmin-2-positive axons, and we found that $96\pm 3\%$ of Schwann cells expressing HDAC8 are associated with Stathmin-2-positive axons. Stathmin-2 (also called SCG-10) is a recognized marker of sensory axons that is expressed in uninjured peripheral nerves and regrowing axons with a higher expression at the growth cone (Shin et al., 2012 and 2014, and Figure S9 of our revised manuscript). We have also added stainings in Figure S12 of HDAC8/Stathmin-2/Neurofilament in ventral roots (fully motor) and sural nerves (fully sensory). These data show no expression of Stathmin-2 in ventral roots, whereas Stathmin-2 is abundantly expressed in sural nerves. In addition, SCs in ventral roots did not express HDAC8, whereas HDAC8 was abundantly expressed in SCs of sural nerves. Interestingly, HDAC8 was also expressed in many axons of ventral roots.

REVIEWER COMMENTS

Reviewer #1 (Remarks to the Author):

I congratulate the authors on the thoroughness of their additions to the manuscript in light of my comments. I am satisfied that they have addressed my concerns to the fullest extent that is practically possible. I am happy to recommend this article for publication.

Two additional comments (these are just for academic debate and should not be added to the manuscript)

1) Apologies with the comment re Babetto 2020 - I agree they didn't show less S6Kinase in their Hif1alpha flox mouse. My mistake.

2) Regarding Jun phosphorylation. Parkinson et al., 2008 showed that Schwann cells from JunAA mice (without the 2 major active phosphorylation sites, Ser 63 and 73) demyelinate like wild type Schwann cells (Fig 6, particularly 6I and J). Obviously this doesn't rule out a role for phosphorylation at other sites Thr 91 and Thr 93. I would disagree with the authors interpretation of figures 3 and 4 in Parkinson et al 2008. In figure 3 Jun Ala (which has all 4 phosphorylation sites inactivated) inhibits Krox20 induction of PZero showing that Jun N terminal phosphorylation is not required for it to act as a negative regulator of myelin genes in cultured Schwann cells. In figure 4 MKK7 expression inhibits Krox20 induction of myelin genes most likely by re-expression of Jun protein (though I accept a role for Jun phosphorylation is not ruled out in this experiment).

We thank very much Reviewer 1 for finding our revisions convincing. We also agree with the additional comments. cJun phosphorylation may indeed not be required for the conversion of Schwann cells into repair Schwann cells, but we understood in Parkinson et al., 2008, that both cJun and phosphorylated cJun can have a similar effect in inducing demyelination and the conversion into repair Schwann cells, if upregulated or overexpressed.

Reviewer #2 (Remarks to the Author):

Hertzog et al., have addressed many of the major concerns and improved the manuscript. However, there are still some minor experimental and editorial adjustments that could further strengthen the study and provide more support to their conclusions.

Comments:

Fig. 1. Thank you for adding before lesion controls to the behavioral tests and adding

another measure for mechanical sensitivity (Von Frey test). As presented, the experimental design and statistical analysis performed is confusing. Were the toe pinch, Von Frey, and rotarod tests performed as between-groups or within-groups experiments? Since the statistics used t- tests for each individual time point, each time point likely is from a different cohort of animals. If that is the case, the line graphs should be converted back to bar graphs. However, if each time point was performed on the same cohort of animals, a repeated measures ANOVA should be performed instead of individual t-tests.

For the Toe pinch and Rotarod tests, we have changed back to bar graphs because not all time points were acquired with the same animal cohorts. For the von Frey test, the same animal cohort was tested at all time points, so we kept the line graph. We hope this solution is acceptable for both Reviewer 1 and Reviewer 2.

We thank Reviewer 2 for the advice to use a two-way mixed ANOVA (repeated measures ANOVA) to identify if the knockout explains mechanical force threshold over time. To strengthen the validity of the results, we additionally calculated linear mixed effect model and Aligned Rank Transform ANOVAS, all showing the same results.

Supplementary Figure 4. Thanks for adding the intraepidermal nerve fiber experiment. Can the authors explain why they used Neurofilament, rather than PGP9.5, an established marker for skin reinnervation assay?

PGP9.5 antibody described in articles of Patrick Enfors' group (e.g. Rinwa et al., Pain, 2021) is unfortunately not available anymore. We thus bought another PGP9.5 antibody (cat. # 14730-1-AP, Proteintech), but it did not give clean enough results to be quantified. Since it is a pan neuronal marker such as Neurofilament M, we used a rabbit anti-Neurofilament M antibody (cat. # AB1987, Merck Millipore), which is very clean and labels neuronal processes very well in the paw samples. Neurofilament, such as PGP9.5, is expressed in all axons and is a specific neuron marker.

Fig. 1&2, Supp 11 and 12. One of the authors main conclusions is that HDAC8 is specifically expressed in sensory SCs and controls SCs repair properties and axon regeneration. However, there are no experiments that directly examine co-localization of HDAC8 with a SCs marker in the nerve. Only close association of HDAC8 positive cells to Stmn2 positive or neurofilament positive axons is shown. While the in vitro experiments and the conditional KO with cre lines are supporting a role of HDAC8 in SC, an in vivo validation that HDAC8 co-localizes with SCs markers around sensory axons and not motor axons should be provided. Such characterization of HDAC8-positive cells could help the reader understand the proportion of SCs expressing HDAC8 in the various nerves.

We have used cross sections to see the tight association of SCs with either myelinated axons or with non-myelinated axons. In the case of myelinated axons, we have used P0 in Supplementary Figure 2 to label specifically myelinating Schwann cells. In the case of non-myelinated axons, we have quantified only nuclei in close contact with an IB4+ axon bundle. However, we also need to have a second neuronal marker in the case of IB4 labeling to make sure that the IB4 signal labels axons. Therefore, we cannot add another channel: we need 4 channels for HDAC8, IB4, NFM and DAPI. In addition, in the past we have tried many times to stain Sox10 in adult sciatic nerves and have never managed to get a specific signal. In unlesioned nerves, Sox10 expression levels are very low (see Fig. S5b), which makes its detection by immunofluorescence very difficult. Nevertheless, we tried again the Sox10 staining with 4 different Sox10 antibodies (Abcam mouse ab216020, Abcam mouse ab212843, DSC Innovative Diagnostik Systeme rabbit S1058C01, rabbit antibody kindly given by Dr. Michael Wegner) and 3 different staining protocols (acetone, Proteinase K, antigen retrieval), but with no success again. We have also tried S100-beta staining (mouse GTX 11179), but in adult nerves it does not appear very specific (it gives good results during postnatal development). As shown in our revised Figure S11 (Figure S12 in first revision), none of the nuclei in ventral roots show HDAC8 expression either in the nuclei or in the cytoplasm around the nuclei. Ventral roots contain only motor fibers. This indicates that Schwann cells surrounding motor axons do not express HDAC8. According to Stierli et al. (2018), 72% of nuclei in adult nerves are Schwann cells. In addition, we found that 96% of Schwann cells expressing HDAC8 surround Stathmin2-positive axons and 4% surround Stathmin2-negative axons. We found that around 60% of Stathmin2-positive myelinated axons are surrounded by HDAC8-positive Schwann cells and around 90% of IB4-positive axons are surrounded by HDAC8-positive Schwann cells. We have now also added the quantification of the percentage of cells expressing HDAC8 in the sciatic nerve and we come up with $54 \pm 6\%$. What we can conclude is that HDAC8 is for sure expressed by a large fraction of Schwann cells surrounding sensory axons but not by Schwann cells surrounding motor axons. The exact percentage of Schwann cells expressing HDAC8 in the sciatic nerve is difficult to calculate, but from Stierli et al, we can deduct that this percentage is 75% at the maximum and 51% at the minimum.

Additionally, images shown in supplementary figures 11 and 12 could be more convincing. Resolution of Figure S11 as presented is not sufficient; there appears to be strong co-localization of HDAC8 with stathmin 2, and the HDAC8 positive cells surrounding these axons is not clear.

We have now removed Figure S11 because this is a repetition of Figure 1g, but we have added the quantification to Figure 1g. There is no co-localization of HDAC8 with Stathmin-2, HDAC8 surrounds Stathmin-2. We can see that very well in Figure 1g, and in revised Figure S11 where we are showing magnifications. In the case of one-to-one

relationships between an axon and a Schwann cell, this is very obvious and in the case of Remak bundles, this is more difficult to see because the Schwann cell processes surrounds small caliber axons. However, we can see that the HDAC8 signal is not co-localizing with the Stathmin-2 signal, however the Schwann cell processes are in tight contact with Remak axons, so the two stainings can appear partially co-localized because of resolution limits. We are showing magnifications of Remak axons in the sural nerve to better show that the HDAC8 signal and the Stathmin-2 signal are distinct.

In Figure S12, ventral root and sural nerve should be shown at the same magnification. Although HDAC8 intensity is highest in the sural nerve, it appears that a fraction of HDAC8 positive cells in the ventral root does not colocalize with axons and stathmin 2 staining does not appear entirely axonal in the sural nerve.

We are now showing the same magnification and same exposure of HDAC8 staining in ventral roots and sural nerves and in addition, we have shown magnifications of some regions in each type of nerve. These stainings show that in ventral roots, there is no nuclei surrounded by HDAC8, therefore no expression of HDAC8 in SCs. We are also now showing an increased exposure of HDAC8 to demonstrate that there is no HDAC8 signal surrounding Schwann cells, even at high exposure. However, in ventral roots, many axons express HDAC8. In addition, there is sometimes some signal that is not colocalized with a SC or an axon, which to our opinion is due to non-specific labeling. In our previous version, we had increased the signal intensity compared to sural nerves to really show that there is no SC expressing HDAC8. Now, we are showing ventral root and sural nerve stainings at the same exposure to avoid confusion. In sural nerves, we now show the NFM staining as single channel and its overlap with the Stathmin-2 staining to demonstrate that the Stathmin-2 staining is specific to axons. In addition, Stathmin-2 staining in ventral root does not label any axon or does not show any strongly labeled structure such as in sural nerves. In the ventral root, there is some faint punctuated signal in between SCs and axons that we also see in sural nerves, which seems to be non-specific signal.

Fig 1f and Supplementary Fig. 10. The authors claim that consistent with faster sensory function recovery, there is an increase in the length of sensory specific STMN2-labeled axons length, but not in the general GAP43-labeled axons. The images provided seem to reflect higher density of GAP43 positive axons past the lesion site in the KO compared to WT. It would be valuable to quantify axon density in addition to axon length, to evaluate the number of axons regrowing in addition to how far they grow, for both Stmn2 and Gap43.

We counted the number of regenerating axons at defined distances from the crush site. We are presenting the results in Fig. 1f for Stathmin-2 and Fig. S10 for GAP43. We found an increased axon density for both STMN2- and GAP43-labelled axons at 1.5 mm

downstream the lesion site in HDA8 KO sciatic nerves compared to controls, however there is a bigger and more significant difference for Stathmin-2-labelled axons. In addition, there is still a significant difference at 2.5 mm distal to the lesion site for Stathmin-2-labelled axons, but there is no significant difference anymore at this distance for GAP43-labelled axons.

Reviewer #3 (Remarks to the Author):

The authors have responded satisfactorily to my review points.

We thank Reviewer 3 very much for finding our revisions convincing.